# Look Who’s Talking: T-Even Phage Lysis Inhibition, the Granddaddy of Virus-Virus Intercellular Communication Research

**DOI:** 10.3390/v11100951

**Published:** 2019-10-16

**Authors:** Stephen T. Abedon

**Affiliations:** Department of Microbiology, The Ohio State University, Mansfield, OH 44906, USA; abedon.1@osu.edu

**Keywords:** arbitrium systems, burst size, latent period, lysis from without, mutual policing, quorum sensing, secondary adsorption, superinfection

## Abstract

That communication can occur between virus-infected cells has been appreciated for nearly as long as has virus molecular biology. The original virus communication process specifically was that seen with T-even bacteriophages—phages T2, T4, and T6—resulting in what was labeled as a lysis inhibition. Another proposed virus communication phenomenon, also seen with T-even phages, can be described as a phage-adsorption-induced *synchronized* lysis-inhibition collapse. Both are mediated by virions that were released from earlier-lysing, phage-infected bacteria. Each may represent ecological responses, in terms of phage lysis timing, to high local densities of phage-infected bacteria, but for lysis inhibition also to locally reduced densities of phage-uninfected bacteria. With lysis inhibition, the outcome is a temporary avoidance of lysis, i.e., a lysis delay, resulting in increased numbers of virions (greater burst size). Synchronized lysis-inhibition collapse, by contrast, is an accelerated lysis which is imposed upon phage-infected bacteria by virions that have been lytically released from other phage-infected bacteria. Here I consider some history of lysis inhibition, its laboratory manifestation, its molecular basis, how it may benefit expressing phages, and its potential ecological role. I discuss as well other, more recently recognized examples of virus-virus intercellular communication.

## 1. Introduction

“…the earliest genetic analyses of phage biology centered on the study of lysis inhibition (LIN)… one of the most venerable problems of molecular genetics…”—Wang et al. [1], pp. 813–814.

Virus-associated intercellular communication has recently been described in the guise of a small-molecule-based mechanism, dubbed arbitrium systems (ASs). These are expressed by some *Bacillus subtilis* phages [2]. A *Vibrio cholerae* phage has also been identified which is able to recognize and respond to a *V. cholerae*-expressed autoinducer, that is, a host quorum-sensing signaling molecule [3] (a phenotype here dubbed, autoinducer-associated prophage induction, or AAPI). In both cases, the result is modification of the timing of phage-induced bacterial lysis. With ASs, lysis is *delayed*, that is, phage lysogenic cycles rather than lytic cycles are more likely to be displayed. With the *V. cholerae* system, lysis instead is *accelerated*, with prophages more likely to be induced. Both mechanisms can be described as examples of phage social behaviors, ones that result in a phenotypic plasticity in the timing of phage lysis [4,5,6,7].

Notwithstanding these newly discovered systems, it was over 70 years ago that phage lysis-timing phenotypic plasticity, as involving intercellular virus-virus communication, was first described, by the combined efforts of Alfred Hershey [8] and Gus Doermann [9]. The parallels between this ‘lysis inhibition’ (LIN) and arbitrium systems (ASs) are striking [10]. Both involve signals that travel from potentially earlier phage-infected bacteria to later phage-infected bacteria. Both involve delays in phage-induced bacterial lysis. Furthermore, both involve, at least in principle, an increase in the number of phages produced by signal-receiving phage-infected bacteria [11,12]. On the other hand, these systems differ in that one is displayed within a context of temperate phages and resulting lysogenic cycles (ASs), while the other is observed particularly with the strictly lytic T-even type myovirus bacteriophages, i.e., phages T2, T4, and T6 [13,14]; see also Gromkova [15] for observation of LIN in T-even-like *Shigella* phages and Schito [16] for observation of LIN in the podovirus N4. In addition, while ASs involve an autoinducer as signal, LIN instead employs whole phage virions.

An additional mechanism of virion-mediated, possible virus-virus intercellular communication involves an acceleration of lysis rather than LIN’s delay. This phenomenon, also as seen with T-even phages [17], was dubbed a lysis-inhibition collapse, and specifically a *synchronized* LIN collapse [18]. The underlying molecular mechanism of synchronized LIN collapse may be based, at least in part, on a phenomenon discovered even earlier than LIN, called lysis from without, or LO [19,20,21,22]. Though synchronized LIN collapse is a phenomenon of lysis acceleration, like autoinducer-associated prophage induction (AAPI), nevertheless synchronized LIN collapse and AAPI are mechanistically dissimilar. Very much like LIN, however, synchronized LIN collapse (i) is a response to a signal originating from other phage infected bacteria, (ii) relies on a signal that consists of intact phage virions, and (iii) provides a response that may be particularly beneficial given an environmental presence of high numbers of phage-infected bacteria.

Building on earlier reviews [20,23], here I consider developments in the study of LIN as well as synchronized LIN collapse, especially as published over the past 25-plus years. Emphasis is placed on a combination of laboratory manifestation, mechanistic bases, and ecology. In the course of the latter, I consider as well the ecology of arbitrium systems (ASs), of autoinducer-associated prophage induction (AAPI), and of yet another, earlier form of virus-virus communication, dubbed here as high-multiplicity lysogeny decisions (HMLDs) [10,24]. We thus now have at least five examples in which bacteriophages may be involved in forms of intercellular communication (Figure 1). See Table 1 and Table 2 for a summary of abbreviations and concepts used in the article.

### 1.1. Communication

Central to the idea that viruses can communicate intercellularly is the very concept of communication. Communication can be a two-way phenomenon, with at least two participants each sending and receiving information to and from the other. A more general concept of communication, however, is that information can be generated by an individual and then received by one or more other individuals, either with or without reciprocation. If the signal is in some manner adaptive, especially as tied in some way to natural selection, then either the generator or the recipient is producing or receiving the information via physiological or behavioral processes and this is towards some benefit to one or both individuals. Such adaptive communication is what is of interest here. Specifically, Diggle et al. [51] reviewed the evolutionary biology of bacterial quorum sensing, differentiating their signals into three types: ‘signals’, coercion, and cues. (Their meaning of ‘signal’ I place in single quotation marks to distinguish this from the less specific meaning of the word, which I leave bare.) These I briefly introduce including in terms of virus-associated intercellular communication.

#### 1.1.1. ‘Signals’

A ‘signal’ benefits both sender and recipient. Quorum sensing thus relies on ‘signals’, with recipients benefiting and responding in a way that also should benefit senders. Likely this is also the case with lysis-delaying arbitrium systems (ASs) among sending and receiving phage-infected bacteria, as ASs essentially are based upon phage-mediated quorum-sensing mechanisms [2].

#### 1.1.2. Coercion

With coercion, the signaler benefits, but the recipient does not. The accelerated lysis of synchronized LIN collapse may be viewed, for example, as a form of coercion. From a different perspective, however, synchronized LIN collapse might also be viewed as involving instead a ‘signal’ as the recipient may be benefiting as well from the signal. The ‘signaling’ and coercion mediators in this case nevertheless are the same thing, free virions, and result in the same consequences, which is lysis of LINed phage-infected bacteria in a synchronized manner. Synchronized LIN collapse, that is, is a more rapid indeed collective reduction in the number of phage-infected bacteria, bacteria which, if left intact, would otherwise be capable of adsorbing and killing virions via expression of superinfection exclusion.

#### 1.1.3. Cues

A cue benefits the recipient of a signal but does not benefit the sender. *V. cholerae* quorum-sensing molecules thus would seem to be cues with regard to prophages that can detect those signals (i.e., via autoinducer-associated prophage induction; AAPI). That is, as bacteria, *V. cholerae* lysogens presumably do not, at least to a first approximation, benefit from their quorum sensing effecting induction of their prophages. Similarly, the induction of lysis inhibition (LIN) or instead high-multiplicity lysogeny decisions (HMLDs) should benefit at least one signal-receiving, bacterium-infecting phage, i.e., the primary phage. The other adsorbing or infecting virions (secondary phages) which stimulate the expression of these phenotypes, however, may solely be supplying an indication of high environmental virion densities (the cue), but may not otherwise participate in the resulting, presumably phage-beneficial lysis delays.

## 2. Overview of Lysis Inhibition

Phage T4 is the best known of the T-even phages [52,53,54], and has been used for nearly all of the modern analyses of LIN and LIN collapse. Early work on LIN involved as well study of phages T2 and T6, however [23]. The phage T4 life cycle under low multiplicity of infection conditions, i.e., as seen with single-step growth experiments [55,56], is typical of a strictly lytic phage [57]. Phage virions thus go through what can be described as an extracellular, diffusion-mediated ‘search’ for bacteria to infect. Upon encounter with a susceptible bacterium, these virions then adsorb (attach) to the bacterium with some probability. That adsorption is followed by translocation of the phage genome into the bacterial cytoplasm. Infection proper then ensues, with infections terminated in conjunction with the release, via lysis, of intracellularly matured virion progeny.

The above-described process is seen with ‘primary’ T4 phages. Here the term primary is not being used in an epidemiological sense but instead in a more biomedical sense [58], that is, to describe the first strictly lytic phage to adsorb and then infect a given bacterium. Subsequently adsorbing phages can be described as secondarily infecting, or superinfecting. As many secondarily adsorbing phages in fact do not infect, i.e., as due to expression of superinfection exclusion by primary phages, I use here instead the terms ‘secondarily adsorbing’ or simply ‘secondary’ phage [18,58]. This terminology is based on the use of “secondarily adsorbed” and “secondary adsorption” by Doermann [9]. That is, T4 phages display mechanisms of superinfection exclusion, notably as expressed by T4 genes *immunity* (*imm*) and *spackle* (*sp*) [23], which have the effect of blocking secondary phage genome translocation and thereby blocking secondary infection, but without blocking secondary adsorption (SA). This is a distinction which is relevant to an appreciation of LIN. Particularly, if ‘superinfection’ is read literally, then assumptions may be made that LIN is a phage-coinfection associated phenomenon, which it is not.

After a brief delay at the beginning of T4 infections, during which T4 gene expression is initiated, the occurrence of subsequent phage adsorption, i.e., of secondary adsorption, induces or indeed stabilizes a delay in the lysis of these bacteria, giving rise to lysis inhibition. The duration of inhibition of lysis that is seen appears to be controlled, at least in part, by the number of virions which secondarily adsorb, with more secondary adsorptions leading to longer LINed latent periods for the adsorbed infections [18,20,23,28]. As the resulting latent period extensions are basically continuations of already ongoing phage lytic cycles, the consequence is prolonged intracellular maturation of new virions. As seen upon eventual lysis, there therefore is an enhancement of the burst size of the primary infections. These burst size enhancements can be quite substantial, i.e., many-fold increases, with LIN-associated latent-period extensions lasting on the order of hours.

The likelihood that a phage-infected bacterium will be secondarily adsorbed is a function of the concentration of free phages found within that bacterium’s environment [29]. The greater the likelihood of secondary adsorption, then the greater the frequency of primary adsorptions, meaning the greater the likelihood of phage infection of any remaining as-yet unadsorbed bacteria. The greater the frequency of primary infections, therefore, then the *lower* the likelihood that phage-uninfected bacteria will remain present within an environment. LIN thus can be viewed as a secondary adsorption-induced mechanism of latent period extension, along with burst-size enhancement, that will tend to occur in association with declines within environments in numbers of phage-available bacteria, as well as increases in numbers of phage-unavailable (already phage-infected) bacteria.

## 3. History of Lysis Inhibition

The concept of LIN, in terms of publication, explicitly dates back at least to Hershey [8]. Aspects of the phenomenon certainly must have been observed prior to that point, as small, cloudy, rough-edged phage plaques, or in terms of long delays prior to the lysis of broth stock preparations, i.e., such as when working with the LIN-encoding phage T2, which was isolated (perhaps) in the 1920s [14]. The real breakthrough in appreciating LIN, however, occurred with Hershey’s study of LIN-defective phages, which he called *r* mutants for ‘rapid lysis’. Hershey, though, pointed to Sertic [59] for possible priority in studying such mutants. Hershey’s discovery was quickly followed by a more-detailed characterization of the LIN phenotype by Doermann [9]. Subsequently, in the 1950s, Seymour Benzer [27] chose rapid-lysis mutants for his much-celebrated studies of the fine structure of genes [60]; see Moussa et al. [36] for a listing of additional *r*-mutant contributions to the development of biology.

### 3.1. Initial Observations of Lysis Inhibition

Hershey [8] described LIN mainly in phage T2 but also phages T4 and T6: (p. 620): “…the mutant… proves to differ from the parent type in causing prompt rather than delayed lysis in undiluted culture…” See Figure 2 for illustration of the wild-type phenotypes as well as a similar phage, RB69, that demonstrates a failure to display LIN. Given single-step growth-type experiments, i.e., which lack in display of LIN due to an avoidance of secondary adsorptions to phage-infected bacteria, T-even latent periods instead typically are roughly 20- to 25-min long whether wild-type [61] or *r* mutant [8]. Hinting at the ecological role of LIN, Hershey also found in broth-serial-transfer experiments that LIN-expressing phages could out-compete an *r* mutant.

The phenotype of particular interest in Hershey’s 1946 publication [8] was that of plaque morphology. The plaques of these *r* mutants Hershey described as having a (p. 623), “…somewhat larger size, which differ [from wild type] in showing nearly complete lysis at the periphery.” That is, wild-type phages that display LIN have smaller plaques which are cloudy in their periphery, what Hershey [8] describes as “a distinct halo of partial lysis”. Typically, wild-type plaques displaying LIN are also described as having rougher edges (less sharp) in comparison with the plaques of *r* mutants [20,27,31]. The rate of spontaneous appearance of *r* mutants was reported as from 10^−3^ to 10^−4^ of a wild-type population [8].

### 3.2. Virus-Virus Intercellular Communication

The 1948 paper by Doermann [9], the findings of which were first reported in outline in 1946 [65], is strikingly similar in approach to that of the Erez et al. [2] discovery of arbitrium systems (ASs). Doermann found especially that LIN was induced by virion adsorption to already phage-infected bacteria. Though the LO phenomenon was discovered about half a decade earlier [66]—and may underlie another form of virus-virus intercellular communication, i.e., synchronized LIN collapse [6,18,21] (Figure 2; Section 5.3)—it was the observation by Doermann [9] that virions released from one phage-infected bacterium can influence the physiology of a second phage-infected bacterium, in a positive manner, that constitutes so far as I am aware the first description of virus-virus intercellular communication. Indeed, again so far as I am aware, Doermann’s [9] observation constitutes the first description also of simply bacterium-to-bacterium intercellular communication. That is, this is communication that is at least ecologically similar to quorum sensing [67], though with whole phages serving as the signaling factor rather than autoinducer molecules.

The question of why phages might display LIN would not be addressed in detail for quite some time past Hershey’s and Doermann’s descriptions. Therefore, that lysis inhibition might represent a mechanism of environmental bacterium-density estimation was not considered until much later [29]. In that publication [29], the question was asked: If the utility of LIN is one of larger burst sizes, then why should the effect be one which is inducible rather than constitutive? Thus, lysis inhibition was suggested as a response to declining numbers of phage-susceptible bacteria within a phage-infection’s environment, i.e., as also suggested by Erez et al. [2] as an explanation for selection for phage arbitrium systems (ASs). In both cases, it is likely that selection for extended latent periods is in response—as an evolutionary algorithm [68], i.e., such as toward maximizing a virus’ basic reproduction number, *R*_0_ [69]—not just to declines in densities of phage-susceptible bacteria but also serves as a response to *increases* in local densities of phage-infected bacteria. The latter especially are bacteria that are capable of inactivating free phages. This inactivation occurs via superinfection exclusion [23] for T-even phages (Section 4.2 and Section 6.3) and via superinfection immunity for temperate phages [70,71,72,73].

Another form of virus-virus intercellular communication mediated by whole phages, high-multiplicity lysogeny decisions (HMLDs), likely also is a response to declining numbers of phage-susceptible bacteria and/or to increasing numbers of phage-restrictive bacteria. Like arbitrium systems (ASs), HMLD is seen with phages which are temperate. In these phages, high multiplicity adsorptions as resulting in coinfections can result in greater likelihoods of display of lysogenic cycles rather than of lytic cycles [48], and this is a phenomenon which has been observed at least since the early 1950s [10]. A common theme of all of these phenomena is the addressing, by phages, of issues that arise in association with high densities of phage-adsorbable and/or -infectable bacteria. These bacteria represent, to phages, either potential phage-infection opportunities while they are still phage uninfected (as presumably thereby more effectively exploited given expression of autoinducer-associated prophage induction; AAPI), as declining opportunities for phage infection of still phage-uninfected bacteria (as presumably relevant to LIN; high-multiplicity lysogeny decisions, HMLDs; and arbitrium systems, ASs), and/or as potential dangers, the latter due to inactivation of free phages by already phage-infected bacteria (as relevant to LIN, HMLD, synchronized LIN collapse, and ASs).

## 4. Lysis Inhibition Laboratory Phenotypes

The basic LIN laboratory phenotypes are those that are seen during plaque formation as well as during undiluted, that is, non-single-step phage infections (for the latter, see Figure 2 and Figure 6). In both cases, perhaps not surprisingly, the differences between wild-type and *r* mutants appear to be due to the lysis characteristics of those phages under conditions where secondary adsorption is likely (again, see Figure 2, with the *r*-mutant-like lysis profile provided there by the LIN-defective phage RB69). In addition, as noted, LIN can result in substantial increases in the burst size associated with infections. In this section I take a closer look at both plaque and broth LIN characteristics, i.e., as observed in the laboratory. Ecological considerations of LIN in planktonic cultures versus biofilms are presented in Section 7.

### 4.1. Plaques

Plaques formed by *r* mutant phages are both larger and clearer than those formed by T4 wild type. What is the explanation for this difference? As a general theme, note that plaque size traditionally has been measured in terms of areas in which bacterial lawn turbidity has been visibly reduced. It, however, presumably is possible for phage virion diffusion as well as phage infection of lawn bacteria to occur without resulting in obvious declines in lawn turbidity. This, in fact, may occur at the very periphery of most phage plaques [74,75]. The same effect may occur but over quite a bit wider an area in association with wild-type T4 plaques, that is, as due to LIN-associated lysis delays. The evidence for this claim—that T4 ‘plaques’ can be larger than literally they appear—comes from three sources: treatment of plaques with chloroform vapor, collisions between T4 wild type and *r*-mutant plaques, and collision of T4 plaques with plaques produced by phage T7.

#### 4.1.1. Plaque Treatment with Chloroform Vapor

Treatment with chloroform vapor can have the effect of lysing unlysed T4 phage-infected bacteria, even those displaying a constitutively extended latent period [76]. This is due to dissolving of the plasma membrane that otherwise would protect the bacterial cell wall from intracellularly located phage lysozyme. Possibly consistently, upon exposure to chloroform vapor the sizes of resulting lawn clearings—or at least the sizes of *clearer* areas (halos) surrounding plaques—become more similar between T4 wild type and *r* mutants versus pre-treatment plaque sizes (Figure 3). This clearing effect Streisinger et al. [77] explained thusly (p. 26): “…halos are due to the action of the phage lysozyme, which diffuses from the plaque and lyses chloroform-killed bacteria surrounding the plaque.” In other words, it is possible that the similar sizes of clearer areas associated with *r*-mutant and wild-type T4 plaques following chloroform-vapor treatment is a function of similar rates of diffusion of the T4 lysozyme protein away from the center of plaques.

Streisinger et al. also noted (p. 26) that, “Many of the ‘halo-less’ mutants produce lysozyme that seems identical to the standard one. These mutants map at locations other than the ‘lysozyme’ mutants… and presumably affect other properties.” Thus, if chloroform-induced halos are not necessarily strictly a function of extracellular lysozyme, then T4 wild-type phage infections instead may fail to lyse sufficiently—until aided by chloroform—to be noticed as part of the visible plaque. This results in wider chloroform-induced halos than induced with the more lysis-proficient *r*-mutant plaques. Indeed, to some degree, under certain conditions, these wild-type ‘halos’ may be noticeable even without chloroform treatment (upper-left image, Figure 3B). Thus, T4 plaque ‘zones of infection’ might be somewhat larger than is traditionally inferred from untreated clearings alone, with wild-type zones of infection perhaps approaching in size those of *r* mutant plaques. 

#### 4.1.2. Colliding T4 Plaques

If it is extracellular lysozyme that is responsible for the larger size of T4 wild-type plaques following chloroform-vapor treatment (Figure 3), then we would expect, for example, that *r* mutant plaques would be readily able to invade the chloroform-clearable areas of a wild-type plaque. That is, extracellular lysozyme alone should not be able to stop ongoing phage adsorption and infection, else plaques themselves would not be able form. Alternatively, if *r* mutant plaques are unable to invade these chloroform-clearable areas of wild-type plaques [65], then this could be due to expression of superinfection exclusion [23] by phage-infected bacteria. If the latter is the case, then it may be possible to infer the actual size of wild-type T4 zones of infection within otherwise untreated lawns. Indeed, this *r*-mutant failure to invade, and resulting potential to infer T4 wild-type plaque size, is exactly what one seems to observe (Figure 4), with again inferred wild-type zone of infection and *r*-mutant plaque size found to be similar in diameter.

#### 4.1.3. T7 Plaques Colliding with T4 Plaques

In an additional experiment, phage T7 was grown on the same plates as both phage T4 wild type and an *r* mutant. Phage T7 produces very large plaques which thereby in principle should be able to envelope the plaques of other phages. At the same time, phage T7 is presumably not able to infect already T4-infected or, perhaps especially, LINed bacteria. The consequence is that enveloping phage T7 plaques may be able to delineate the borders of phage influence (presumed zone of infection) associated with wild-type T4 plaques. Upon performing such an experiment, the resulting area of influence of T4 wild-type plaques turns out to be similar to those of an r mutant, and more so where T7 has spread to these areas last given that T7 plaques grow faster than T4 plaques (Figure 5).

#### 4.1.4. Wild-Type T4 Plaques, Conclusions

The above-described experiments do not prove that phage T4 is infecting bacteria somewhat outside of what is the wild-type visible plaque, but they nevertheless are suggestive that this is what is occurring. Based on that premise, we thus can consider that phage T4 wild-type and *r* mutants propagate outward at similar rates from plaque initiation points towards what ultimately will be ‘plaque’ or at least ‘zone of infection’ peripheries. Presumably this occurs, for both phage types, via a standard ‘reaction-diffusion’ type propagation mechanism [74], that is, where especially rapid lysis-type infections (single-step growth-length latent periods) are followed by virion diffusion and then infection of new cells. Without LIN, then this is simply the way that phage plaques form. With LIN, by contrast, these rapid-lysis-length infections are presumably so-lysing predominantly at extreme plaque or zones of infection peripheries, and doing so spatially ahead of secondary adsorptions. Behind these infections and thereby closer to the center of plaques, however, then secondary adsorptions occur with greater likelihood, and so too should LIN. These now LINed phage-infected bacteria do not rapidly lyse, resulting in substantial turbidity beyond the visible periphery of these “plaques”, despite ongoing ‘zone of infection’ growth. This relative lack of lysis occurs to such a degree that traditionally we may not even consider that these zones of infection are a part of the plaque proper. From Abedon and Yin [81], we can further speculate that the existence of bacteria within bacterial lawns as microcolonies, rather than as randomly dispersed individual cells, can contribute to the LIN reduced-rate-of-lawn-lysis effect. See [80,82,83] for consideration of phage penetration into bacterial microcolonies more generally, and Section 7.2 for discussion of analogous dynamics as potentially could occur during phage T4 exploitation of bacterial biofilms. 

### 4.2. Broth-Growth Aspects

There are basically three ways to manipulate a broth culture to display LIN. (i) The most conceptually straightforward though not necessarily always the most effective means is to infect a majority of bacteria with phages, wait some span of time such as 15 min (during which primary infections are actively metabolizing), and then adsorb a second dosage of phages to a majority of the phage-infected bacteria [31]. Ratios of added phages to bacteria of approximately 5 can be employed for both steps. Bacteria must be supplied at sufficiently high densities, such as 10^8^/mL, that this ratio of 5-to-1 results in fairly rapid bacterial adsorption by phages [84]. (ii) It is also possible to skip the explicit secondary adsorption step, allowing instead non-simultaneous adsorptions along with resulting non-synchronized lysis to supply secondarily adsorbing phages to those phage-infected bacteria which have not yet lysed (see Figure 2). With this second approach, however, substantial culture lysis in some cases may occur prior to the induction of LIN, resulting in less dramatic LIN-associated increases in turbidity and thereby less readily observed intervals over which lysis of these cultures eventually occurs [18]. (iii) LINed cultures also can be established with ratios of phages to bacteria of somewhat less than one (e.g., 0.1), along with reasonably high bacterial densities, such as 10^7^ or 10^8^/mL [6,18,29] (consider starting with a fewer bacteria if LINed cultures subsequently end up reaching too high turbidities such that subsequent LIN collapse is inefficient). The initially added phages will adsorb and infect bacteria, assemble new phages until the end of a normal (rapid lysis) latent period, and then lyse. The released phages will infect most of the remaining bacteria found within the broth culture, some of which will subsequently lyse after another rapid-lysis type latent period. The latter lysis will not be well coordinated, however, and thereby can rapidly give rise to sufficient numbers of secondarily adsorbing phages that LIN is robustly induced in a majority of phage-infected bacteria found within these cultures. See Figure 6 as an example, especially in the vicinity of as labeled, “Start of lysis inhibition”.

Ultimately, given sufficient nutrient and/or oxygen densities [18] (speculation for the latter), then cultures will display an abrupt drop in turbidity. This drop is a *synchronized* lysis-inhibition collapse, as considered in greater detail below (Section 5.3, Section 5.4, and Section 5.5; see also Figure 2 and Figure 6). More relevant at this juncture is that the delay in lysis of LINed phage-infected bacteria, i.e., as prior to the start of LIN collapse, can continue for hours. During this time, the LINed bacteria making up these cultures remain virion absorbable (Figure 6). As the infected bacteria presumably are displaying superinfection exclusion [21,23], the result is that these cultures can represent hostile environments to free phages [6]. This then provides a second explanation for Hershey’s [8] observation that LIN-displaying wild-type phages can out-compete *r* mutants in broth culture: Not only does LIN result eventually in greater burst sizes for wild-type phages, but so too earlier lysing phages—as definitively seen with rapid lysing *r* mutants—will likely be negatively affected at high rates due to adsorption by their released virions to LINed bacteria. This phenomenon actually creates a dilemma for the not-lysed wild-type bacteria: To lyse? (Thereby resulting in released virions adsorbing to previously infected bacteria and thereby being inactivated.) Or instead to never to lyse? That issue is considered further below in terms of synchronized LIN collapse. First, though, I review what is known of the molecular mechanisms underlying the lysis delay of LIN.

## 5. Mechanisms

“Thus, at last, the outlines of a molecular explanation for the principal sacrament of the Phage Church are apparent, but many questions remain.”—Wang and Young [85], p. 113.

Multiple genes and thereby multiple gene products have been associated with phage T4 lysis and LIN. In this section, I provide overviews of these players and their roles in both LIN as well as LIN collapse, the latter synchronized as well as unsynchronized.

### 5.1. Overview of Key Players

To understand the mechanisms underlying expression of LIN, it is crucial to first appreciate those underlying rapid lysis. Rapid lysis occurs at the end of a standard phage lytic cycle and is triggered by what can be described as a lysis from within (LI, contrasting with both lysis from without, i.e., LO, and LIN). For tailed phages, LI can involve the action of at least three phage proteins: a holin, a lysozyme or endolysin, and a spanin [86,87]; the latter (spanin) will not be considered here. Endolysins are cell-wall digesting enzymes that are responsible for observable phage-induced bacterial lysis. These, however, also are not the primary consideration here. Rather, it is the holin that is most relevant to LIN as it controls the timing of both phage-induced LI and initiation of the cell-wall digesting activity of lysozyme [20,85,86,88,89,90]. To block LIN, it is therefore logical that the action of the T4 holin protein must be inhibited.

Holins are phage proteins that act by forming holes in the plasma membranes of bacteria [1,91,92]. These holes both disrupt the plasma membrane’s chemiosmotic potential and allow lysozyme to digest the surrounding cell wall. The T4 phage holin is encoded by the T4 *t* gene [93]. The endolysin, in turn, is encoded by gene *e*. Though phage T4 LI can be understood, more or less [86], in terms of gene products T and E [20], LIN requires as well consideration of various phage T4 *r* genes and their gene products [94]. Gene *rI* has been the focus of the most attention in terms of the mechanism of LIN. Genes *rIIA* and *rIIB*, described as “cistrons” as standing for “contiguous segments” [27], seem to give rise to an overcoming of LIN by an otherwise unrelated mechanism [20,30]. They will not be considered here, though historically are the *r* genes studied by Benzer [27,60]. Gene *rIII* can be important to LIN expression as well, serving as a second inhibitor of protein T [38,92], a so-called antiholin.

Gene *rIV* is gene *spackle* (*sp*) [94], which otherwise is gene 61.3 of phage T4 [95]. Gene *sp* along with gene *immunity* (*imm*) will be considered while covering LIN collapse (Section 5.3, Section 5.4, and Section 5.5), as their roles may have more to do with resisting LIN collapse than inhibiting protein T. In addition is gene *5*, which plays a role that is associated with that of gene *sp*. Sp essentially serves as a foil to the action of the 5 protein, where the latter is provided by secondarily adsorbing phages. Imm also seems to serve as a foil, in this case to the cell-damaging actions of secondary adsorptions.

Gene *rV* is identical to gene *t*, and various gene *t* mutations exist which are defective in LIN [31,94]. These mutations presumably interfere with the signal transduction pathway that is propagated from secondary adsorption events to T-holin inhibition. In addition are genes *rI.-1* and *rI.1* [96] which may play some as yet undefined role in phage T4 display of lysis inhibition.

### 5.2. Inhibition of T-Holin

“Somehow, LIN is established by a signal transduction process in which the attempted injection of the superinfecting phage DNA is detected and relayed to the lysis system by products of the famous phage *r* genes.”—Wang et al. [1], p. 813.

The blocking of the hole-forming action of T holin directly results in a blocking of LI and thereby serves as the mechanistic underpinning of LIN. A working hypothesis for the induction of LIN is that secondarily adsorbing phages provide some signal, e.g., periplasmic DNA or associated phage proteins [30,34], which in some manner is transduced, essentially instantaneously [30,97], to plasma-membrane-located protein T (perhaps ‘instantaneous’ because otherwise unstable RI protein, below, is already T bound). The result, as mechanistically summarized in Figure 7, is inhibition of T-hole formation, thereby giving rise to LIN.

At a minimum, the RI protein thus likely plays a role in secondary adsorption signal transduction to protein T [30,32], while the RIII protein along with RI together serve to stabilize the not-yet hole-forming form of protein T. Though not necessarily directly related to LIN, both RI and RIII also may be important to T4-infection productivity (burst size) given very slow bacterial host growth [98]. It is conceivable that the T4 Sp and/or Imm proteins also could play some role in propagation of the secondary adsorption signal [38] (Section 5.4). Though T protein is inhibited from forming holes during LIN, addition of cyanide ion, a metabolic poison, results in an artificially induced lysis (from within) including of LINed bacteria [32], that is, unless protein T is constitutively not functional [76]. As follows, I consider the roles of these various molecular players in LIN expression in greater detail. 

#### 5.2.1. Protein T

Gene *t* is expressed during T4 infections as a late gene [31], and accumulates in the infected cell inner/plasma membrane, a barrier which must be breached for the T4 lysozyme to reach the bacterial cell wall. To prevent T-hole formation, and thereby to prevent both the metabolic poisoning of the infected bacterium and lysozyme from reaching the cell wall, T-hole formation must be inhibited. As noted, inhibition begins with some as-yet-unidentified signal associated with phage secondary adsorptions, and is otherwise associated with the T4 RI protein interacting with the T holin prior to T-hole formation.

#### 5.2.2. Protein RI

RI was found to be a periplasmic protein [30,33,37,99], one which initially is tethered to the plasma membrane [34]. Upon release from this tethering, RI is highly labile within the periplasm and thus degrades rapidly, at least in terms of its active conformation [34]. Secondary adsorptions in some manner lead to a stabilization of RI, generating what Dressman and Drake [31] described as RI*. RI* can serve to specifically stoichiometrically (1:1) inhibit T-hole formation, thereby delaying LI [36]. This inhibition is accomplished perhaps simply via some form of stearic interference with initiation of T–T interactions [32]. Engineered so that it is a less labile, then RI instead can accumulate in the periplasm and consequently inhibit T protein even without the secondary adsorption signal [33]. See also the results of Los et al. [100].

These RI properties suggest that it is inhibition of RI degradation that is the third step of LIN expression, i.e., in order, (i) secondary adsorption is followed by (ii) generation of the secondary adsorption signal, which (iii) prevents RI degradation, and only then (iv) can binding of adequate amounts of RI protein to T protein (v) prevent the formation of T-holin-mediated holes in the plasma membrane. The T4 RI protein thus can be viewed as a phage secondary-adsorption activatable antiholin.

Gene *rI* may also contribute to longer phage latent periods under single-step growth (not LINed) conditions in a non-T4 genetic background. Upon serial-transfer evolution of phage RB69, which is a T-even-like phage that is unable to display LIN (Figure 2), a number of phage mutants exhibiting shorter-than-parental latent periods were isolated [17]. A majority of these mutants, indeed six of six sequenced, displayed *rI* mutations, a total of five different mutations among them, while only two of these six mutants harbored gene *t* mutations [6]. This is circumstantial evidence that the RI protein might display an antiholin function in a not-strictly T4 phage, i.e., as resulting in slightly shorter phage latent periods in its absence. Indeed, it may be easier mutationally to acquire shorter latent periods as a competitive strategy by knocking out a gene function, such as that of an antiholin, rather than to “subtly modify a function”, such as towards generating a gene *t* “clock” mutant [99].

#### 5.2.3. Protein RIII

An antiholin function for the RIII protein [32] was molecularly corroborated by Chen and Young [38]. Unusual for an antiholin, RIII appears to act on the cytoplasmic rather than periplasmic side of the plasma-membrane localized T protein. Thus, inhibition of LI during LIN is a function of the binding to T holin of two T4 proteins, RI and RIII, though only RI is thought to be responsible for transducing the secondary adsorption signal. Indeed, Chen and Young suggest that RIII acts as a stabilizer of the RI-T complex which is required for T holin inhibition during LIN. 

#### 5.2.4. Genes rI.-1 and rI.1

Possible additional contributions to the repertoire of genetic functions contributing to LIN are considered by Golec et al. [96]. Genes *rI.-1* and *rI.1* together with the *rI* gene (but not gene *rIII*) make up a single operon. What Golec et al. did was to express these genes in *E. coli*, either at lower or higher levels (the latter, ‘over-expression’), in various combinations from plasmids. They then infected the so-expressing bacteria with either phage T4 wild-type (T4D) or a phage T4 *rIII* mutant. Cultures were then followed by turbidity (lysis profile). Toward increasing the clarity of the following discussion of their results, which otherwise can be somewhat complex due to the number combinations of factors involved, I differentiate distinct experimental results by using capital letters, i.e., A through M. See Table 3 for a summary of the experiments discussed.

With phage T4 wild-type infections, it was found (A) that genes *rI.-1* and *rI.1* over-expressed from a plasmid could each slightly extend the LINed latent period as determined by lysis profile. (B) Over-expression of gene *rIII*, by contrast, could greatly extend this otherwise wild-type LINed latent period, while (C) over-expression of gene *rI* had little if any impact. These results are suggestive that (C) having more RI protein present, particularly at the beginning of infections—as gene expression from plasmids is expected to terminate early during phage T4 infections—can have little impact on the stabilization and therefore extension of LIN. By contrast, (B) having excess RIII would appear to stabilize LIN beyond the normal timing of initiation of LIN collapse, the latter an issue that I return to in Section 5.3.2. This RIII stabilizing effect may be consistent with the suggestion by Chen and Young [38] that RIII may stabilize the RI-T complex, though alternatively this RIII result might instead suggest a nonspecific effect which only indirectly results in delays in the timing of LIN collapse.

In combining the over-expression of these various genes, of particular note is (D) the impact on LIN duration of combining the over-expression of genes *rI* and *rIII* versus gene *rIII* over-expression alone. That is, the combination, unlike gene *rIII* alone, only slightly extended the timing of initiation of LIN collapse beyond that of T4 wild-type with no cloned-gene expression. In addition, (E) is a substantial positive impact on the duration of LIN of combining over-expression of genes *rI and rI.1*. It is difficult to say with any confidence what these latter results might imply, however, as the experiments, as above, were done in an otherwise wild-type-phage genetic background. Nevertheless, these results are suggestive that (D) excess RI-protein presence might interfere with a LIN- and/or RI-T complex-stabilizing impact of excess RIII protein. That is, the RIII protein if present in excess might be able to contribute to extending the duration of the LINed latent period (e.g., by excessively stabilizing the T-holin or RI-T complex), but with that effect in some manner countered by the over-expression of the *rI* gene early during infections. In addition, (E) gene *rI.1* might have some sort of stabilizing impact on protein RI.

Subsequent experiments by Golec et al. [96] were done based on lysis profiles of an *rIII*-mutant phage. Not surprisingly, assuming reasonable stability of the RIII protein (i.e., as contrasting the stability of RI), (F) gene *rIII* over-expression complemented the phage *rIII* defect, resulting in an extended LIN display as equivalent to that seen with gene *rIII* over-expression in association with the wild-type phage infection. Of interest, however, (G) over-expression of genes *rI.1* and *rIII* in combination, given infection with the *rIII* phage mutant, seems to have resulted in only slightly longer LIN duration than those infection lengths seen with wild-type phage infections without additional phage gene expression. This result compares with that of (F) over-expression of gene *rIII* alone with the same, *rIII*-minus, genetic background, which results in the noted greatly extended LIN duration. (H) Interference is possibly also seen with over-expression of a combination of genes *rI.-1* and *rIII*, though the level of interference is comparatively slight. (E) Over-expressed gene *rI.1* thus may contribute, in combination with over-expression of *rI* antiholin, to extended wild-type phage LIN duration, but (G) also detract from the ability of over-expressed gene *rIII* to overly complement a phage-associated defect in gene *rIII*.

A challenge with these experiments and their interpretation involves (I) a toxicity associated with expression of gene *rI.1* from a plasmid, necessitating delaying induction of expression of this gene to simultaneous with phage addition, but which then would be expected to be followed soon after with a phage-mediated termination of that expression. (J) The toxic effect of gene *rI.1* expression, however, could be suppressed by co-expression of gene *rIII*, suggesting some sort of association between these genes or resulting protein products. This proposed association between these two genes or their products perhaps would be equivalent in origin to (G) the negative impact that gene *rI.1* over-expression has on the ability of over-expressed gene *rIII* to overly complement a phage *rIII* defect with regard to LIN duration. At the same time, (K) gene *rI* over-expression suppressed (thus resulting in presence of toxicity to bacteria) this (J) gene *rIII* suppression of gene *rI.1*-associated toxicity (as resulting instead in an absence of toxicity to bacteria), doing so perhaps in some manner that is mechanistically equivalent to the (E) positive impact of over-expression of genes *rI.1* and *rI* on LIN duration with a wild-type phage infection background.

A yet further complication is that (L) gene *rIII* over-expression alone, without phage infection, was found to slow down the growth of expressing bacteria. This slowing, however, (M) was not seen when gene *rI.-1* was co-over-expressed with gene *rIII* (nor with gene *rI.1* co-over-expression), a result that is perhaps similar in its mechanistic basis to (J) the ability of gene *rIII* expression to suppress the toxicity associated with gene *rI.1* expression without phage infection. My primary general conclusions from these results are that they are suggestive of an interplay between various genes or protein products in determining the timing of initiation of LIN collapse, and furthermore that gene *rIII* in particular potentially may have a substantial impact on delaying LIN collapse. I return especially to the observation (B and F) of gene *rIII* over-expression-associated extended LIN duration in the Section 5.3.2. 

### 5.3. Lysis-Inhibition Collapse and Its Synchronization

“…the subversion of the LIN state in *e sp* or *e 5* infections may simply reflect LO brought about by the baseplate lysozyme of the secondary phage running amok.”—Young [20], p. 449.

To disseminate as virions, phages must release those virions from the bacterium that they are infecting. Thus, in order to be successful viruses, LINed phage infections eventually must lyse. We can speculate that mechanistically this lysis might represent simply a failure at some point of continued RI- and RIII-mediated inhibition of LI (mechanism 1 as discussed below; Figure 8; Section 5.3.1). However, is that the whole story? There is at least one reason to think not, and that is the propensity for LINed T4-infected broth cultures to lyse *less* rapidly in the absence of continued secondary adsorptions, particularly in the absence of secondary adsorptions that otherwise would have occurred later in LINed cultures [18]. In addition, LINed cultures lyse *more* rapidly given the supplying of substantial numbers of additional secondarily adsorbing virions, e.g., given explicit secondary adsorptions at multiplicities of 50 or 70 [18]. Thus, the paradigm that more secondary adsorption results in more LIN [28] may be overly simplistic. At the same time, however, a realistic alternative mechanism underpinning the timing especially of the *initiation* of LIN collapse is not yet available. Phage T4 also seems to possess an additional route to LIN collapse, that resulting from drops in temperature [101]—thus hinting at an ecological utility seen during infection exiting from colonic environments—but this mechanism has been little studied and is not covered further here.

#### 5.3.1. Four Mechanisms Potentially Leading to Lysis-Inhibition Collapse

The reason that the occurrence of more secondary adsorptions does not necessarily result in longer delays until LIN collapse may be a consequence of there existing two contrasting forces governing the length of the latent periods displayed by LINed bacteria. The first is maintenance of inhibition of T-hole formation as mediated by the activated RI protein and presumably by RIII protein as well (Section 5.2.2; Figure 7; failure of this maintenance is mechanism 1, below). The second is failure of maintenance of bacterial cell-envelope integrity (mechanisms 2 through 4, below). From these considerations, I thus present four mechanisms, in Figure 8 and immediately below, which may contribute to the lysis of LINed phage-infected bacteria and their cultures, i.e., LIN collapse.

Recall that LIN collapse may be differentiated into two distinct phenomena, which are described here as synchronized versus unsynchronized LIN collapse (Figure 2), where mechanisms 1 and 2 would contribute predominantly to unsynchronized LIN collapse, and indeed initiation of LIN collapse, whereas mechanisms 3 and 4 would contribute predominantly to synchronized LIN collapse, and thus LIN collapse’s subsequent progression. In addition, only the latter, synchronized LIN collapse, is hypothesized to represent a mechanism of virus-virus intercellular communication, whether by coercion or instead by ‘signal’ (Section 1.1.2).

The four mechanisms possibly contributing to LIN collapse are:Lysis from within (LI). RI- and RIII-mediated inhibition of T-holin ceases and normal LI therefore commences. Perhaps secondary adsorption-associated RI stabilization in the periplasm can only last so long [34] and is not otherwise replenishable over the long term during LIN.Membrane deterioration (MD). Eventually the plasma membrane of phage T4-infected bacteria becomes unstable, resulting in metabolic poisoning of the LINed bacterium and thereby a triggering of T-hole formation, i.e., “nonspecific deterioration of the membrane” [20] (‘membrane deterioration’; MD). LI thus commences. I consider this mechanism to be possible but nevertheless somewhat hypothetical (see also Section 5.3.2).Lysis from without (LO). In the course of continued secondary adsorption, cell walls become sufficiently degraded that LO commences. This results in plasma membrane disruption, metabolic poisoning of the phage-infected bacterium, and thereby LI as well.Secondary traumatization (ST). In the course of continued secondary adsorption, plasma membranes become sufficiently degraded as to trigger T-hole formation, thus, as above, with LI commencing. I consider this mechanism also to be somewhat hypothetical, though perhaps less hypothetical than mechanism 2. In the case of mechanisms 1 and 2, the result is an initiation of LIN collapse that may be equivalent to a lysis from within (LI) of LINed phage infections. Across a culture of phage-infected bacteria, these two mechanisms likely can give rise to only an unsynchronized LIN collapse (Figure 2). Reception of the released-virion signal by still intact LINed bacteria appears to initiate mechanisms 3 and 4 by processes seemingly more akin to lysis from without (LO) than LI, though like mechanisms 1 or 2, with release as well of virion extracellular signal. The additional extracellular signal appears to give rise to an acceleration and/or more widespread occurrence of mechanisms 3 or 4, resulting in a synchronized LIN collapse.

As indicated, mechanism 4 might result from something equivalent to what may be described as secondary traumatization (ST). From Cornett [102]: “The turbidity and viability measurements seen with *imm*^-^ superinfected cells indicate that superinfecting phage did not cause extensive [LO] in these cells but appeared to ‘traumatize’ the *imm*^-^-infected cells, thereby reducing their ability to form infective centers and to continue protein synthesis. The cause of the observed traumatization is not known…” That is, this would be the killing of phage-infected bacteria from without (as due to excessive secondary phage adsorption), but killing by means that is other than via lysis from without (LO) [102]. ST in turn might be viewed as a secondary adsorption-caused “deterioration” of the plasma membrane. Note in any case that distinguishing among mechanisms contributing to LIN collapse is complicated by the involvement of secondary adsorption also in the expression of LIN (Section 5.2; Figure 7). These various mechanisms of LIN collapse are further phenomenologically dissected in Table 4.

#### 5.3.2. Initiation of LIN Collapse: Ruling out Mechanism 2?

The results of Golec et al. [96], as discussed in Section 5.2.4, are suggestive of a role of holin stabilization in determining the duration of LIN, that is, in determining the timing especially of the initiation of LIN collapse. As noted (Section 5.2.3), the RIII protein in LIN has been found to serve as a cytoplasm-side preventer of T-hole formation [32,38]. As the presence of gene *rIII* over-expression appears to result in somewhat longer LINed latent periods [96], this is suggestive that the timing of initiation of LIN collapse is determined more internally to LINed phage infections, i.e., as via mechanisms 1 and 2, rather than externally.

Specifically, an intact plasma membrane is presumably required for this proposed RIII protein-associated holin-stabilizing effect, and resulting delay in the start of LIN collapse. A role for gene *rIII* over expression in delaying the initiation of LIN collapse therefore would seem to rule out mechanism 2, at least for LIN collapse as begins absent gene *rIII* over-expression. That is, if the RIII protein can extend LINed latent periods by acting presumably solely on T holin, then the associated plasma membrane must not otherwise be deteriorating independent of T-holin action. Thus, if membrane deterioration (MD) does not initiate LIN collapse when excess RIII protein is present, then presumably MD does not initiate LIN collapse when excess RIII protein is *not* present, i.e., as during wild-type phage display of LIN. Perhaps then we can return to mechanism 1 as underlying the timing of the start of LIN collapse, with subsequent synchronization of LIN collapse then determined by mechanisms 3 or 4. This hypothesis as to the contributions of mechanism 1 versus mechanism 2 to the initiation of LIN collapse, however, is dependent on the impact of gene *rIII* over-expression [96] *not* being a consequence of *nonspecific* RIII impacts—such as on phage infection metabolism more generally—and thus the RIII protein acting in some nonspecific manner towards inhibition of MD. Nevertheless, the Golec et al. [96] results are suggestive that RIII protein in excess can serve to delay LIN collapse in phage T4-infected bacteria.

#### 5.3.3. Timing of Initiation of Lysis-Inhibition Collapse versus Its Synchronization

LIN collapse, as effected by mechanisms 1 or 2 (lysis from within or, instead, membrane destabilization giving rise to lysis from within), may be distinguishable from LIN collapse as effected by mechanisms 3 or 4 (lysis from without or secondary traumatization). These mechanisms can be viewed as resulting in less synchronization of LIN collapse (“unsynchronized” LIN collapse; mechanisms 1 and 2) versus more synchronization (“synchronized” LIN collapse; mechanisms 3 and 4), respectively. This latter distinction appears to be in terms of the impact of continued secondary adsorptions on the kinetics (synchronized LIN collapse) rather than necessarily the timing of the start of LIN collapse (at which point only unsynchronized LIN collapse is observed; Figure 2). That is, respectively, this is the occurrence of more synchronized *progression* of culture-wide lysis of LINed phage-infected bacteria [18] rather than modifications of the timing of LIN collapse’s less synchronized *initiation*.

Particularly, addition of anti-T4 serum soon after LIN induction can result in an LIN collapse that begins only marginally *later* than without blocks on secondary adsorption. Indeed, the length of these latent periods is essentially identical to what is seen if antiserum addition is delayed until just prior to synchronized LIN collapse (Figure 4 of [18]). In addition, the timing of initiation of LIN collapse can vary little when using conditionally adsorption-incompetent phages, i.e., phages which when released are incapable of secondarily adsorbing (Figure 5 of [18]). Thus, the actual timing of initiation of LIN collapse, once LIN itself has been fully initiated, may be somewhat independent of further secondary adsorption. That is, it can be shown to occur 100 or more min after anti-T4 serum addition (thus 100-min or more post the last secondary adsorptions) as well as occur with infections that are otherwise incapable of releasing adsorption-competent virions. These observations would appear to be more or less consistent with mechanisms 1 or 2 controlling the timing of initiation of LIN collapse, while the previous Section (5.3.2) provides an argument supporting especially mechanism 1 as underlying this timing of the start of LIN collapse.

Regardless of the degree to which the timing of initiation of LIN collapse is *not* a function of the extent of secondary adsorption, once LIN collapse has been initiated, then this collapse may, as noted, be accelerated via secondary adsorption. Across LINed cultures, the result is a synchronization (more rapid) versus an unsynchronized (more gradual) LIN collapse [18]. Indeed, the resulting synchronized lysis resembles LO. Unlike LO as traditionally considered [22], however, phage infections displaying LIN should be full of cytoplasmic E lysozyme [103], indeed even more full of E protein than phage-infected bacteria just prior to rapid lysis [93]. Thus, not only may secondary adsorptions lead to lysis by disrupting cell walls (i.e., via the action of 5 protein associated with secondary phages and thereby LO/mechanism 3), but so too such adsorption might disrupt plasma membranes more directly (secondary traumatization, i.e., ST/mechanism 4), resulting in either case in metabolic poisoning and thereby LI.

### 5.4. The Phage Imm and Sp Proteins

Genes *imm* and *sp* are expressed by primary infecting T4 phages and serve to interfere with the damaging impacts of adsorptions by other, secondarily adsorbing T4 phage virions [23]. Within a context of LIN, the resulting phenotypes might collectively be described as a resistance to premature LIN collapse [21]. The Sp protein is thought to localize in the periplasm of T4-infected bacteria [95] and it interferes especially with LO [102]. LO appears to be effected by the 5 protein, a cell-wall digesting enzyme as displayed by adsorbing T4 phages [104,105]. Too much cell wall digestion presumably is the underlying basis of LO, which Sp serves to inhibit in already phage T4-infected bacteria. Indeed, a T4 gene *5* mutation exists which seems to allow the resulting virions to bypass this protective Sp action [106].

Though the Imm protein reportedly molecularly resembles a holin [91], its action nevertheless is less easily understood than Sp’s. Perhaps it interferes with the ability of secondary adsorptions to disrupt the plasma membrane of phage-infected bacteria, as may be described as a resistance to the above-noted secondary traumatization, ST [102]. Consistently, Imm appears to be a plasma membrane protein [107]. Indeed, ST might be initiated at the point of phage tail tube-tip interaction with the plasma membrane as would otherwise be towards effecting secondary phage DNA translocation into the bacterial cytoplasm [108]. To the extent that such disruption could either trigger the action of otherwise inhibited T-holin, or allow E lysozyme to bypass T in gaining access to cell walls from within, then conceivably an absence of Imm, like defects in Sp, could result in a LO-like response to secondary adsorptions. Indeed, *imm* mutants are partly defective in resisting LO [102]. Possibly consistent with these ideas, the application after induction of LIN of anti-T4 serum to *imm*-mutant-infected cultures results in LINed latent periods which are nearly as long as seen in wild-type cultures absent antiserum addition, whereas without antiserum addition these *imm* mutant latent periods can be shorter than as seen with T4 wild type by roughly one half [21]. At a minimum, therefore, the presence of Imm protein in some manner seems to delay the initiation of LIN collapse, and we can speculate that it does so by preventing damage that results from secondary adsorptions, thereby perhaps inhibiting mechanism 4, above.

Mutated gene *sp* [95,102,106,109], like gene *imm* [21], also gives rise to a partially defective LIN, with *sp* mutants somewhat more defective than *imm* mutants [21,102]. By defective, I mean that lysis of LINed bacteria is observed sooner than with wild-type T4, and this can be described as a *premature* LIN collapse. These effects might be reminiscent of what Hershey [65] described as weak inhibitors of LIN, particularly perhaps for what are now known as gene *imm* mutants, but see also Burch et al. [94] for alternative explanations for that weak inhibition (Hershey also identified “stronger” than wild type lysis inhibitors). By contrast, *sp* mutants are nearly fully defective in their display of LIN in some experiments [95,109] but not in others [21,102], and perhaps are defective particularly at higher temperatures, possibly indicating that Emrich’s [109] *sp* mutant is temperature sensitive [21]; see also Burch et al. [94]. In addition, genes *imm* and *sp* may be additive in their effects as *imm sp* double mutants appear to be completely deficient in displaying LIN under conditions where neither *imm* nor *sp* mutants are also completely deficient [21,102].

Are resulting LIN defects due to reduced resistance to LO [110], or equivalent (e.g., secondary traumatization; ST), or instead due to some other mechanism, i.e., such as reduced outright induction of LIN? Unfortunately, the answer to this question is not certain. My suspicion, as based on the experiments presented in Abedon [21], is that a presence of either *imm* or *sp* is necessary for LIN induction, but the presence of both is not necessarily required under all circumstances. Meanwhile, once LIN has begun, something equivalent to higher sensitivity to either LO (gene *sp* mutants) or plasma membrane disruption of E-containing bacteria (gene *imm* mutants and ST) presumably gives rise to early LIN-associated lysis (premature LIN collapse) in single *sp* or *imm* mutants (mechanisms 3 and 4, respectively). Clearly, however, additional experimentation is needed, especially to determine whether *imm sp* double mutants are truly defective in transducing the inhibition of T upon secondary adsorption, versus *imm sp* double mutants simply being highly sensitive to disruption by secondary adsorption.

### 5.5. Lysis Inhibition and Unsynchronized as Well as Synchronized Lysis-Inhibition Collapse

Based on the various ideas considered in this section (Section 5) and the previous (Section 6), we can consider a timeline of mechanisms underlying LIN-associated phage infection cycles:LIN induction is associated with secondary adsorption as it transduces a signal which allows periplasmic RI protein to interfere with or continue to interfere with T-hole formation. This secondary adsorption may result in some degree of cell envelope damage but the degree of damage is reduced due to the actions of the Imm and Sp proteins.Lysis of a fraction of individual LINed bacteria, especially near the point of initiation of LIN collapse (unsynchronized LIN collapse) and perhaps as caused by erosions of R-protein mediated inhibition to T-hole formation (mechanism 1), results in a buildup of phage virions within the environment.The presence of these additional extracellular (free) virions results in an accumulation of secondary adsorptions of still-intact LINed phage-infected bacteria.At some point, rates of bacterial lysis accelerate as a direct consequence of increasing numbers of secondary adsorptions (mechanisms 3 or 4).Additional lysis gives rise to further buildups of secondary adsorptions of remaining intact phage-infected bacteria, providing a positive-feedback lysis, i.e., synchronized LIN collapse. Lysis inhibition, as a product of virus-virus intercellular communication, thus leads to an unsynchronized LIN collapse which in turn becomes synchronized across a population of LINed phage-infected bacteria, with the latter potentially also a product of virus-virus communication. The following three sections consider phenotypes resulting from these and other forms of virus-associated intercellular communication from more ecological perspectives.

## 6. Evolutionary Ecology of Lysis Inhibition and Synchronized Lysis-Inhibition Collapse

Evolutionary ecology is the study of the *why* of adaptations. For our purposes, here it is a question of why display an inducibly longer latent period (LIN) rather than solely rapid lysis. In addition, the utility of synchronized LIN collapse is considered. I begin, however, by considering the possible utility of rapid lysis.

### 6.1. Utility of Rapid Lysis

The default state of all strictly lytic phages is some approximation of rapid lysis, even if latent periods are constitutively long. Rapid lysis, as defined here, that is, is simply a phage lytic cycle that does not result in LIN, nor result in some other inducible deviation from a standard-length phage latent period. Notwithstanding that definition, however, the question of why display rapid lysis, as being posed here, is one of why *not* instead display a constitutive LIN? That is, why not display somewhat longer strictly lytic latent periods all the time?

An answer to this question is that extended phage latent periods, such as many hours long versus many tens of minutes long—even given resulting enhancements in burst size—are not optimal for phage population growth under all circumstances [4,5,17,111,112,113,114,115,116,117]. Those circumstances where longer latent periods may be especially disfavored are when relatively high concentrations of phage-uninfected host bacteria are present in combination with head-to-head (same culture) competition with shorter latent-period phages [17,35]. This disadvantage, if these conditions hold, will stem from shorter latent-period phages reaching and infecting more bacteria sooner than can longer latent period phages, because phages with longer latent periods spend more time infecting bacteria rather than diffusing as virions to infect new bacteria.

To achieve this shorter latent-period advantage, then the longer latent period of competitor phages must be constitutive rather than inducible. That is, this proposed head-to-head competition cannot simply be between one phage which *cannot* display LIN (such as an *r* mutant) and another phage which, once more or less all bacteria have become phage infected, *can* then come to display LIN (i.e., an inducible longer latent period, as displayed by T4 wild-type phages). The competition instead has to be between a constitutively shorter latent-period phage and a constitutively longer latent-period phage where neither can also display LIN. What then specifically might be the advantages to a so-competing phage of an inducibly longer latent period such as LIN? 

### 6.2. Utility of an Inducibly Longer Latent Period (Lysis Inhibition)

There are at least two explanations for the utility of LIN, and these correspond to circumstances of low densities of phage-uninfected bacteria, on the one hand, and circumstances of high densities of phage-infected bacteria on the other. Note, however, that these two circumstances are not necessarily mutually exclusive. 

#### 6.2.1. Reduced Densities of Phage-Uninfected Bacteria

“…presumably to avoid further dispersal of progeny into a host-deficient environment.”—Wang et al. [1], p. 813.

The utility of LIN presumably is that it represents an *inducibly* longer latent period. In the absence of induction, phages thus should be able to take advantage of faster population growth as potentially obtained by shorter latent periods (Section 7). The utility of shorter latent periods is found, however, particularly when phage-uninfected bacteria are abundant in an environment (Section 6.1). 

Longer latent periods, alternatively, can be advantageous given a relative dearth of phage-infectable bacteria, that is, when the cost of displaying a longer latent period is less high. In particular, if fewer bacteria are present, then the phage latent period takes up a smaller fraction of the overall phage generation time. That generation time consists of (i) a period during which phages are finding bacteria to infect—a period which is longer the fewer phage-sensitive bacteria that are present—and then (ii) a period during which infection is occurring (latent period). Thus, if it takes, for example, many hours on average for a free phage to find a bacterium to infect, then there can be utility to displaying much larger burst sizes even if displaying larger burst sizes is accomplished at the expense of extending phage latent periods by a few hours.

Despite such potential utility to longer latent periods when phage-uninfected bacteria are relatively scarce, it is only when phages are first replicating within cultures that are relatively dense with target bacteria that this advantage of LIN may be realized, since otherwise there will be insufficient numbers of phages produced to secondarily adsorb and thereby induce LIN. Under these latter circumstances, phages will display a shorter latent-period phenotype until phage numbers come to exceed bacterial numbers. Because bacterial densities are relatively high, a majority will then become phage infected, but so too will phage-infected bacteria become secondarily adsorbed with relatively high probability, thereby inducing LIN. LIN as an inducible longer phage latent period therefore is an adaptation, at least in part, to environments in which relatively high densities of phage-uninfected bacteria are declining in numbers, and particularly declining in numbers explicitly as a consequence of substantial amounts of phage infection [29].

#### 6.2.2. Increased Densities of Phage-Infected Bacteria

There is a second utility to LIN than simply larger burst sizes [29]. As noted, T-even-type phage infections display superinfection exclusion, which kills secondarily adsorbing T-even-type phages. The longer an infection lasts, then the longer that free-phage killing ability will persist. LINed bacteria thus can be formidable free phage-killing entities, and this killing can be especially substantial when these phage-infected bacteria are present at high densities, i.e., just as one would expect following growth of populations of phages in the presence of relatively high densities of target bacteria (Figure 6). The second utility of LIN thus is that by not lysing, then associated phage virions remain intracellular, a location that protects them from adsorbing to LINed bacteria.

This utility to not lysing nevertheless results in a problem, indeed a dilemma. That is, a phage infection cannot disseminate its virion progeny without lysing. Lysing amid LINed bacteria however is to some degree suicidal with regard to the released virions (Figure 6). This means that there may be little incentive for a LINed phage-infected bacterium to lyse, or at least to lyse first. If no phage infections lyse first, however, then all phage infections should at least *attempt* to avoid lysing. This precisely is a cost of a constitutively shorter latent period phenotype in the midst of LINed phage infections, and one reason that *r* mutants should be expected to be selected against during broth-culture phage propagation [8], i.e., as they cannot help but lyse earlier than LINed phage infected bacteria. There is, though, a way around this dilemma, at least for wild-type LINed bacteria, and that is for *all* LINed bacteria to lyse ‘first’. This result in fact is approximated by the phenomenon described above as a synchronized LIN collapse (Section 5.3), which at least arguably represents another form of T4-associated virus-virus intercellular communication.

### 6.3. Utility of Synchronized Lysis-Inhibition Collapse

Synchronized LIN collapse may be beneficial to both earlier- and later-lysing LINed phage infections. This benefit should derive from a reduced likelihood of suicidal secondary adsorptions by lysis-released virions as numbers of still intact phage-infected bacteria are reduced, with the faster the depletion in those numbers then the greater the benefit. Secondarily adsorbing virions thus may be viewed as coercing a more rapid lysis of still-intact LINed infections, or from a more biosociological perspective, effecting a ‘mutual policing’ [118,119] against ‘defecting’ (i.e., free phage-killing) not-yet-lysed phage infections [6]. Especially to the extent that this coercion is efficient across an LINed culture, resulting in many rather than few accelerated lysis events, then the coercedly lysed phage infections should benefit as well. Indeed, later lysing coerced phage-infected bacteria should benefit more than earlier lysing (coercing) phage infections given the resulting shorter time frame of the former’s exposure to not-yet lysed phage-infected bacteria.

Though potentially beneficial, it nonetheless is not known whether synchronized LIN collapse represents an evolutionary adaptation. That is, it is uncertain whether phage genes are present which could vary in such a way as to increase or decrease the likelihood of LINed bacteria lysing in response to substantial numbers of secondary adsorptions. Synchronized LIN collapse nevertheless can still be viewed as adaptive, i.e., as resulting in an increased evolutionary fitness for participating individuals, whether lysed or to-be-lysed bacteria, just not necessarily an exclusive product of evolution acting to optimize the synchronization of the lysis of LINed cultures. Rather, mechanisms 3 and 4 (Section 5.3.1) could instead be results of optimization of virion adsorptive processes (i.e., phage penetration of the bacterial cell wall and plasma membrane) or lysis from within (LI), the latter as may be potentiated given secondary adsorption-mediated cell wall and/or plasma membrane disruption (Section 5.3.3). On the other hand, it seems possible that genes *imm* or *sp* could vary allelically in such a way as to subtlety modify an LINed bacterium’s susceptibility to secondary adsorptions. Furthermore, given mechanism 1 as a possible determinant of the timing of initiation of LIN collapse, it would seem that allelic variation could exist that affects the persistence of interactions between R and T proteins over the course of LIN, i.e., thereby modifying the timing of the start of LIN collapse if not necessarily its subsequent synchronization. 

It is of further interest, from an evolutionary ecological perspective, that LINed phage-infected bacteria seem to lyse even without ongoing secondary adsorption, just somewhat less rapidly (Section 5.3). This suggests that LIN collapse synchronization is tunable. That is, when a LINed phage-infected bacterium does not find itself surrounded by other LIN-expressing bacteria, then lysis may be more gradual due to a relative absence of secondarily adsorbing phages, and so too the danger to releasing phage virion progeny may be lower. In addition, speculatively, it is possible that intracellular virion maturation may be continuing as well over these extended lysis times given a lack of synchronized LIN collapse, ultimately resulting in yet larger burst sizes. By contrast, when LINed bacteria are densely present within an immediately local environment (e.g., close together, within micrometers), then a synchronized LIN collapse may be more likely to occur, and this is at the same time that the risk to releasing virion progeny unilaterally should also be highest. Thus, uncrowded conditions might be more effectively exploited by lysis-inhibited phage infections given less-synchronized LIN collapse whereas crowded conditions might be more effectively exploited given more-synchronized LIN collapse.

## 7. Ecology of Lysis Inhibition

We can posit reasons for phage T4 to display LIN in terms of virus-virus interactions, such as contributing to a phage’s competitive abilities (Section 6). Nevertheless, what is the advantage of LIN in a broader ecological context? This question I address from a perspective of less spatially structured (planktonic) environments or more spatially structured environments, the latter particularly as found in association with bacterial biofilms. Discussions of how these scenarios may be manifest as laboratory phenomena, particularly during broth-culture or within-plaque phage population growth, respectively, are presented in Section 4.

### 7.1. Ecology of Lysis Inhibition among Planktonic Bacteria

In fluid environments, LIN supplies phages with larger burst sizes and particularly does so when the displaying of longer latent periods by individual phage-infected bacteria is no longer as costly, that is, less costly in terms of rates over which phages are able to find new bacteria to infect (Section 6.2.1). With more phages produced, then the potential for survival of at least one virion from a single phage burst is increased, as so too is the potential for at least one released phage per burst to eventually find a new bacterium to infect. Since LIN is expected to be displayed especially when phage-susceptible bacteria are in decline, we might expect that any bacteria to be reached by this LIN-associated increased number of released phages will be located some distance away from the lysing phage infection and/or some relatively long span of time into the future. Under both circumstances—that is, large distances between host populations or delays in recovery of local host populations post phage-induced population-wide lysis—then the time until free virion adsorption of a new bacterium may be expected to be long relative to the length of a LINed phage latent period.

This idea of utilities to LIN based on increased burst sizes given increasing separation of bacteria in either space or time is perhaps best illustrated in terms of phage-biofilm interactions (Section 7.2), a perspective based in part on ideas of phage-bacteria interactions during phage-plaque growth [80,120]. Specifically, with bacterial growth within biofilm there can be substantial spatial separation between populations of bacteria (as between individual biofilms) without substantial spatial separation of bacteria as found within such populations (as within individual biofilms). This is the same model as supplied in the previous paragraph, except that bacteria are now found within spatially localized populations due to growth as biofilms rather than due to separation of populations of phage-susceptible planktonic bacteria by some other means.

### 7.2. Ecology of Lysis Inhibition among Biofilm Bacteria

Biofilms, or individual single-species bacterial microcolonies making up biofilms, represent bacteria existing in what can be described as a clumped dispersion. This contrasts with the random dispersion of bacteria as can be found given planktonic existence.

Within clumps, bacteria of the same phage-susceptibility type can exist at rather high local densities. At the same time, substantial distances between clumps should result in potentially long durations of between-bacteria virion diffusion. The high local density of bacteria making up clumps presumably would be conducive to LIN expression, as well as give rise to locally high densities of phage-*infected* bacteria. Locally, high densities of phage-infected bacteria could select evolutionarily for both the extended latent periods of LIN (Section 6.2.2) and subsequent synchronized LIN collapse (Section 6.3). The lower bacterial densities found outside of clumps could, in turn, also select evolutionarily for LIN, but in this case especially for the associated larger burst sizes (Section 6.2.1).

In this model of phage-mediated exploitation of bacterial biofilms, a limited number of virions would initiate a ‘focus’ of exploitation (focus of infection; ‘1′ in Figure 9). A thus spatially isolated phage infection will inherently display rapid lysis since, with relative isolation, then secondary adsorption would be less likely. Multiple surrounding phage-susceptible bacteria found within a single clump, surrounding that initial phage infection, would then relatively quickly become infected by the progeny virions it released (‘2′ in Figure 9). Others of these newly released virions by chance will diffuse further, infecting, e.g., adjacent or further microcolonies (‘3′ and ‘4′ in Figure 9). More locally acquired bacteria, such as those found in the same microcolony as a lysing infection, would be more likely to be secondarily adsorbed as well, either multiply from the same phage burst or instead due to becoming surrounded by other phage-infected bacteria [83], resulting in both display of and selection for LIN. More distantly acquired bacteria, as found at the peripheries of growing infection foci, would by contrast be less likely to be secondarily adsorbed, or be surrounded by phage-infected bacteria, resulting in both display of, and selection for, instead a lack of LIN expression.

The more bacteria infected, and the more phages produced per infected bacterium, then the more phages produced per focus of phage infection (as found within the green band in Figure 9). Thus, with an ability to display an inducible LIN, phages within infection foci can as a group exploit two otherwise conflicting ecological strategies. These are ones of more rapid local acquisition of new bacteria to infect (shorter latent periods, i.e., without display of LIN), as would occur at the growing peripheries of infection foci (e.g., as represented by ‘4’ in Figure 9), versus ones of boosted virion production per bacterium acquired (larger burst sizes, i.e., with display of LIN), as would occur more centrally within infection foci (e.g., as represented by ‘2’ in Figure 9). Together these strategies would give rise to an enhanced exploration of both more-local and more-distant bacteria to infect, the latter such as new biofilms which could support new phage foci of infection (‘5′ and ‘6a’ in Figure 9) [80,121,122,123]. At least in terms of modulating their burst sizes and latent periods, strictly lytic phages which cannot display LIN instead must compromise between such enhancement of local explorations versus more distant explorations for new bacteria to infect.

## 8. Other Virus-Associated Communication Mechanisms

LIN appears to be an adaptation to a combination of both declining densities of phage-susceptible bacteria (Section 6.2.1) and increasing densities of phage-infected bacteria, that is, for the latter, bacteria that can kill adsorbing virions (Section 6.2.2). Synchronized LIN collapse could be a phage response to high densities of phage-restrictive bacteria, particularly to high densities of LINed phage infections which have not yet lysed (Section 6.3). Shorter phage latent periods may be viewed as adaptations especially to higher densities of phage-susceptible bacteria (Section 6.1) while longer latent periods instead can be adaptations especially to dearths of phage-susceptible bacteria (Section 7). What then ecologically may be the utilities to phages of high-multiplicity lysogeny decisions (HMLDs), arbitrium systems (ASs), and autoinducer-associated prophage induction (AAPI)? These I consider in this section.

### 8.1. High-Multiplicity Lysogeny Decisions

With high-multiplicity lysogeny decisions (HMLDs), lysogenic cycles are more likely given the occurrence of high-multiplicity phage infection of a bacterium [10,47,48,124,125]. Associated with high-multiplicity phage infections should also be declines in numbers of phage-susceptible bacteria [126] as well as increases in the prevalence of bacterial lysogens carrying homoimmune prophages, with the latter, via superinfection immunity, being antagonistic to secondarily adsorbing phages. Lysogeny resulting from HMLDs therefore may be viewed, analogous to LIN [6,11], both as a form of potentially protective lysis delay (re: given high prevalence of homoimmune lysogens) *and* as a means of producing greater numbers of new phages. The latter would be a consequence of a combination of lysogen binary fission and subsequent prophage induction in multiple cells, as relevant especially to when phage-uninfected bacteria are declining in number, i.e., as due to the noted high-multiplicity phage infection of bacterial populations. Not all infections, even given HMLDs, will succeed in lysogenizing, however, even if lysogenic cycles versus lytic cycles would appear to be beneficial under a given set of circumstances.

We also can question whether HMLDs are a product of natural selection, versus simply a consequence of inherent kinetics of high-multiplicity versus single-infecting-phage gene expression. That is, in the absence of alleles specifically underlying a phenotype’s expression (HMLDs), and only that phenotype’s expression (versus underlying also lytic-lysogeny decisions during infection by only a single phage), it can be difficult to conclude that a given phenotype evolved for the sake especially of that phenotype (a similar argument may be made for synchronized LIN collapse; Section 6.3). Thus, while HMLDs apparently could be useful to temperate phages as a form of virus-virus communication, it is not obvious that HMLDs may be unequivocally described as phage adaptations.

### 8.2. Arbitrium Systems

Arbitrium systems (ASs) essentially represent phage-encoded quorum-sensing systems [2,127,128]. Nevertheless, generally the same arguments apply as for high-multiplicity lysogeny decisions (HMLDs), except that ASs presumably would seem to be unquestionably products of natural selection. A perhaps relevant question [10], however, is why ASs are even needed if HMLDs are also possible? Perhaps circumstances can exist where infection in the vicinity of existing lysogens, as initiated by the same phage type [129], is of higher likelihood than high-multiplicity infection. Furthermore, perhaps the two systems are in some manner additive in their effects, where high-multiplicity adsorption in the vicinity of existing lysogens (again, of the same phage type) provides even higher likelihoods of lysogenic infection than either HMLDs or ASs acting alone. Indeed, the canonical AS on its own only bestowed about 50% lysogenic cycles, suggesting potential room for improvement in that rate [2].

### 8.3. Autoinducer-Associated Prophage Induction

Contrasting the other four mechanisms of phage-associated communication (Figure 1), autoinducer-associated prophage induction (AAPI) [3,130,131] represents an outlier. Perhaps, however, this is to be expected given that it represents a form of bacteria-to-virus rather than virus-to-virus intercellular communication. In particular, AAPI gives rise to an acceleration of lysis rather than a delay, with this acceleration propagated endogenously rather than being externally imposed (i.e., rather than as seen with synchronized LIN collapse). Furthermore, one has to consider just what might be the ecological circumstances where quorum sensing would rationally serve to stimulate prophage induction [24]. Especially, this would have to be where a phage-infection’s goals must be ones of simultaneously (i) giving up their lysogenic cycle, (ii) presumably seeking as virions new bacteria to infect, and (iii) doing these things without being killed upon secondarily adsorbing especially to fellow homoimmune, prophage-carrying lysogens, i.e., as would display superinfection immunity against the newly released virions.

From the synchronized LIN collapse scenario, we can speculate that were prophage induction and resulting lysis to occur simultaneously among spatially associated sister lysogens, then—assuming also a utility to prophage induction—such autoinducer-mediated stimulation of prophage induction might make sense [24]. Alternatively, we just might not have enough information. For example, perhaps this mechanism is more about prophage induction as a means of eliminating competitor bacteria [50,132,133,134,135], as colorfully dubbed ‘kill the relatives’ by Paul [136], or instead for biasing early lytic-lysogeny decisions towards lytic cycles [24], and in both cases therefore rather than selected for the sake of phage virion dissemination from lysogens. It is noteworthy also that a phenotypically equivalent response to autoinducers, in this case acyl-homoserine lactones, has been found as well among soil- and groundwater-associated temperate phages as well as a phage λ lysogen of *E. coli* [49].

## 9. Conclusions

Communication between viruses as mediated by extracellular factors has been recently described in the guise of the newly discovered phage arbitrium systems. These new systems are of interest molecularly as they provide novel takes on quorum sensing, autoinducer detection, and phage lytic-lysogeny [2] as well as prophage induction [3] decisions. They also help to tell us something about social interactions as they can occur even among phages. Such interactions are even broader than as discussed in some depth here, however, including not just lysis inhibition, synchronized lysis-inhibition collapse, high-multiplicity lysogeny decisions, arbitrium systems, and autoinducer-associated prophage induction, but also superinfection exclusion [23], superinfection immunity [70,71,72,73], phage-expressed anti-unrelated-phage resistance systems [137,138,139,140], crosstalk between prophages during polylysogeny [141,142], exchange of genes between different phage types [143,144,145,146], and phages parasitizing other phages [147]. The study of the social lives of viruses is thus both quite new and at the same time somewhat well established, with lysis inhibition having played a prominent early role in its development.

## Figures and Tables

**Figure 1 viruses-11-00951-f001:**
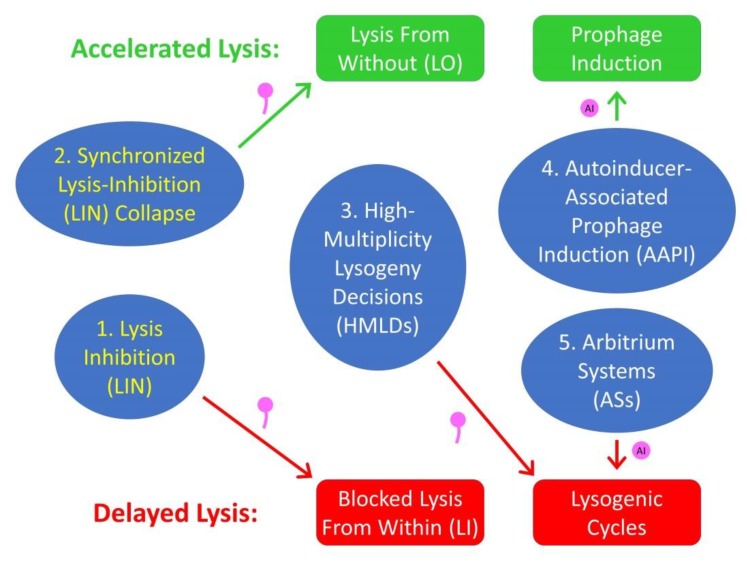
Virus-associated mechanisms of intercellular communication. These can be differentiated as resulting in either accelerated or delayed lysis. The two phenomena indicated to the left are (**1**) lysis inhibition (LIN) [6,8,9,18,20,23,25,26,27,28,29,30,31,32,33,34,35,36,37,38] (for early work discussing genes involved, see also as cited in [23]) and (**2**) synchronized LIN collapse [6,18,21,23], both of which are phenotypes emphasized here. LIN, synchronized LIN collapse, and (**3**) high-multiplicity lysogeny decisions (HMLDs) are all secondary-adsorption, or in the case of HMLD, also secondary-infection associated phenomena [39,40,41,42,43,44,45,46,47,48]. By contrast are (**4**) autoinducer-associated prophage induction (AAPI) [3] (see also [24,49,50]) and (**5**) arbitrium systems (ASs) [2] (see also [10,50]), where instead small-molecule autoinducers serve as effecting signals). Note in addition that while LIN, synchronized LIN collapse, HMLDs, and ASs are all at least arguably examples of virus-virus intercellular communication, AAPI instead represents instead a form of bacterium-to-virus intercellular communication.

**Figure 2 viruses-11-00951-f002:**
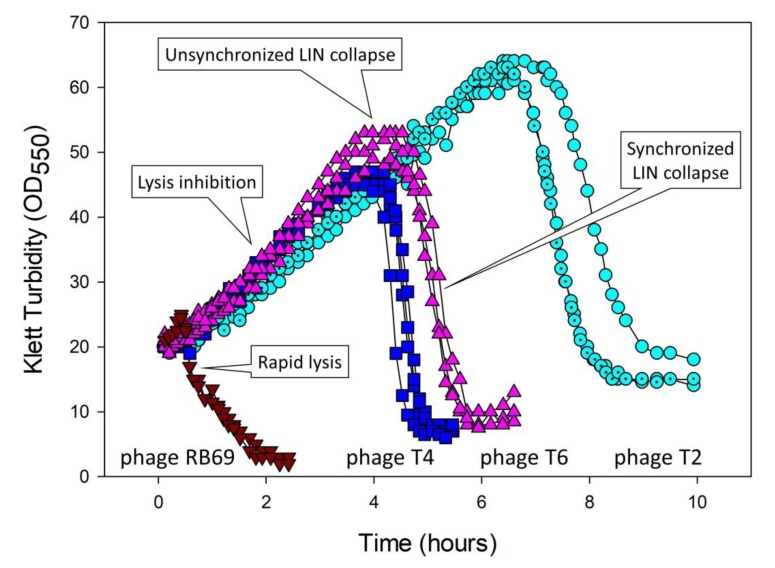
Examples of phage myovirus lysis profiles. Shown, left-to-right, are phage RB69 (brown, inverted triangles, and which does not display LIN but instead rapid lysis) along with the LINed phages T4 (blue squares), T6 (pink triangles), and T2 (aqua circles), with each curve generated from a phage stock obtained from a different source. The three stocks of phage T2 were designated as simply T2 (leftward circles), T2L (circles with center dots, also found to the left), and T2H (rightward circles). *E. coli* CR63 was employed as the host, with an initial density of approximately 10^8^ colony-forming units/mL. Phages were applied only once, at time zero, with a multiplicity of 10. A similar lysis profile to that of phage RB69, but by the phage T4 *rI* mutation, *r*48, is published in Paddison et al. [30]. Synchronized LIN collapse for phages T2, T4, and T6 is observed as rapid, that is, steep declines in culture turbidities. This contrasts with an unsynchronized LIN collapse, which can be observed just prior to the start of synchronized LIN collapse. Unsynchronized LIN collapse can instead occur over longer periods by adding anti-virion agents such as anti-virion serum to an LINed culture prior to or even following the start of their LIN collapse, or by initiating LINed cultures using phages which are unable to release intact virions [18] (these latter experiments are not shown in the figure). With only unsynchronized LIN collapse, culture-wide lysis thus can occur over the course of an hour or more, and this is rather than the minutes as seen here especially with phages T4 and T6, where LIN collapse instead is synchronized [18] (the longer duration of LIN collapse observed with phage T2 in the figure I speculate could be due to depletion of nutrients in the cultures stemming from the long span of phage T2’s LIN prior to that collapse, as addition of nutrients, specifically casamino acids, to highly turbid T4 cultures during broth stock preparation can result in more efficient lysis of those stocks [62]). Note that the observed rise in turbidity following phage application is *not* due to significant bacterial division, i.e., as considered further below (Figure 6), but instead presumably is due to increases in the turbidity associated with individual phage-infected bacteria [63,64]. This experiment was originally published in Abedon et al. [17], with the figure redrawn for presentation here.

**Figure 3 viruses-11-00951-f003:**
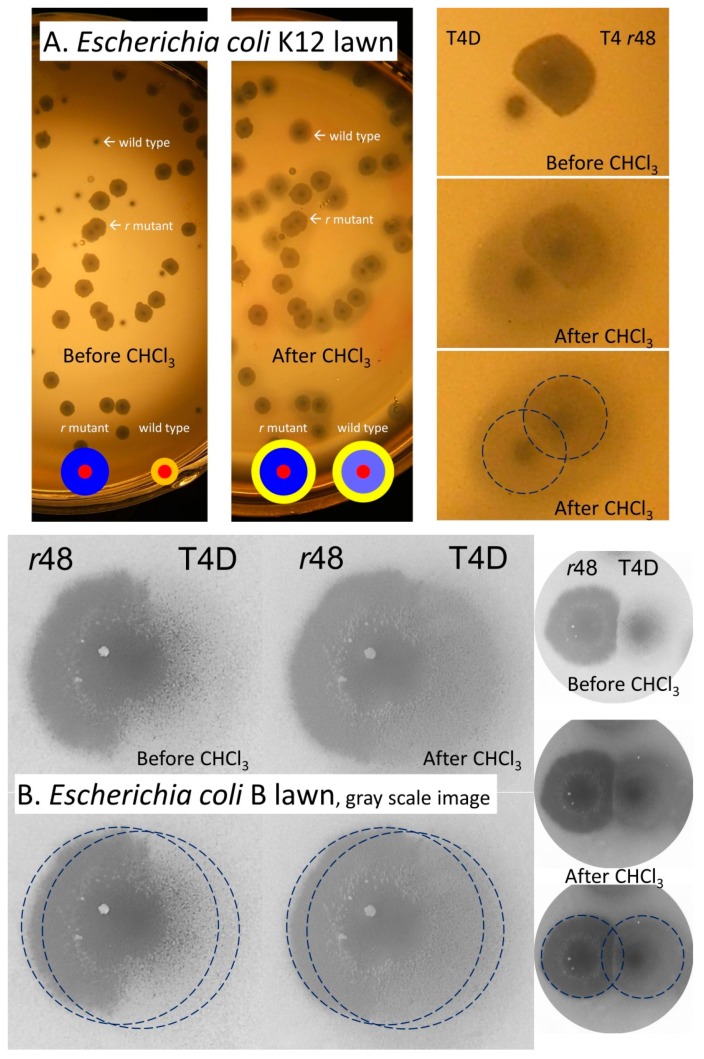
Phage T4 plaques before and after exposure to chloroform vapor. Panel (**A**) is with an *E. coli* K12 lawn and panel (**B**) is with an *E. coli* B lawn. Both show various individual phage T4 plaques growing within standard-sized (100 mm) Petri dishes. Dashed circles are of identical sizes within individual sub-panel figures, drawn to approximate the size of the associated *r*-plaque. These circles are provided to allow more facile comparisons between the wild-type and *r* mutant plaques following chloroform treatment. Phage *r*48 is a gene *rI* T4 mutant and T4D is simply a wild-type strain of phage T4. The T4D plaques are clearly smaller than the *r*48 plaques prior to chloroform-vapor treatment, as labeled in panel A to the upper-left, but similar, at least in size though not in clarity, following chloroform treatment. Plaques were generated and chloroform-vapor treated by Cameron Thomas-Abedon, working under my supervision. A schematic is provided at the bottom of panel A, to the left, showing an approximation of what is being seen. Red indicates the well-defined, least turbid center as seen with both wild type and *r* mutant, presumably indicating mostly complete lysis of the bacterial lawn in both cases (complete lysis that possibly is a consequence particularly of phage interaction with lawn bacteria early in lawn development). Orange indicates, as drawn surrounding untreated phage wild-type plaques, what likely is inefficiently lysed lawn, as seen prior to chloroform treatment (perhaps Hershey’s [8] “distinct halo of partial lysis”). Yellow indicates a poorly defined plaque exterior that becomes emphasized via chloroform-vapor treatment. This is possibly associated with the impact of extracellularly located lysozyme, i.e., as discussed by Streisinger et al. [77]. Blue represents only minimally turbid plaque regions as seen both before and after chloroform treatment (as indicated as well by the various dashed circles found in the figure). This region presumably is associated with the extent of zones of infection whether by wild-type phage plaques, as visualized only after chloroform treatment, or as seen both before and after chloroform treatment with the *r* mutant. The blue region for wild-type T4 phages, that is, presumably represents a region consisting of substantial numbers of LINed phage-infected bacteria (zone of infection) but which retain a lawn-like appearance prior to chloroform-vapor treatment because those bacteria are still largely intact despite being phage infected (and thus this region is not indicated in the pre-chloroform-treatment wild-type plaque schematic as shown to the left). After chloroform treatment, however, these LINed bacteria appear to have become lysed, as indicated by a substantial reduction in turbidity. This lysed region, though, is shown as a lighter shade of blue than as for the *r*-mutant plaques since reductions in turbidity for the wild-type plaques do not appear to be quite as extensive as is seen in the equivalent region with *r* mutant plaques prior to chloroform treatment. These experiments are otherwise unpublished but in part represent visualizations of some of the observations of Streisinger et al. [77].

**Figure 4 viruses-11-00951-f004:**
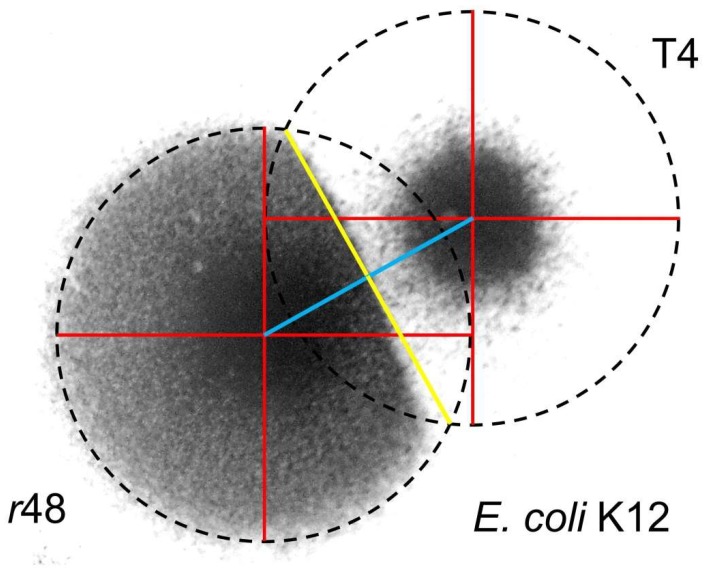
Colliding wild-type and *r*-mutant plaques, *without* chloroform treatment. The circles in this image are of identical size as too are the red lines, which cross at approximately the middle of each plaque. A yellow line is drawn from the intersections of the two circles, and a blue line is drawn from center-to-center of the two plaques. The intersection of those two lines (yellow and blue) coincides more or less with the limit of phage *r*48 plaque growth, presumably as limited by the presence of superinfection exclusion- expressing T4 wild-type phage-infected bacteria, which would be found on the T4 side of the yellow line. Our prediction would be that the *r*48 plaque would have grown to approximately the diameter of the circle in all directions in the absence of interaction with the T4 plaque. Equivalent colliding-plaque observations can be seen in the plaque photographs presented by Hershey and Rotman [78], which are based on phage T2H and *E. coli* strain S (which I speculate is a smooth derivative of *E. coli* B); see also Hershey [65] as well as Lanni [79]. The conclusion from this experiment is that the zone of infection for the phage T4 *r*48 mutant and phage T4 wild-type are similar on this *E. coli* K12 host, else the yellow line would not so closely track the periphery of *r*48 plaque growth when in close proximity to a T4 plaque. Plaques were generated by Cameron Thomas-Abedon and these experiments are otherwise unpublished.

**Figure 5 viruses-11-00951-f005:**
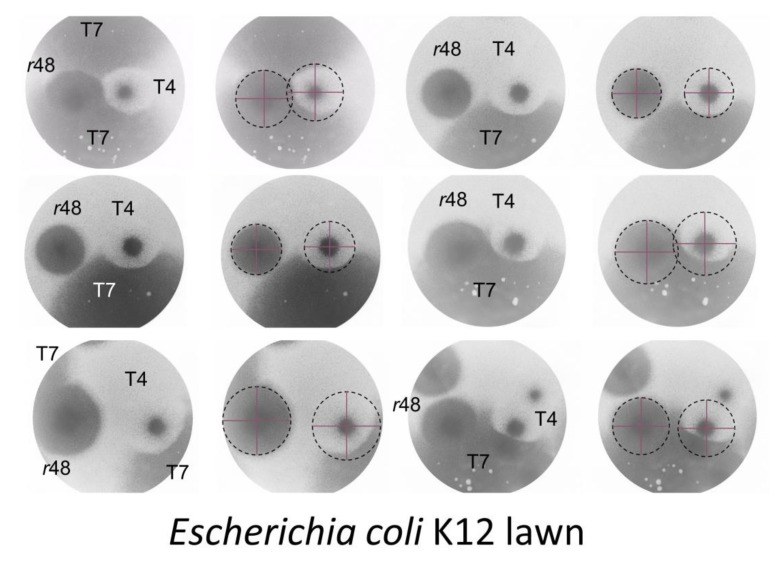
Various examples of collisions between individual plaques (not spots) of phage T4 wild type, the phage T4 *r*48 mutant, and phage T7. Six different examples are shown, with the first and second columns identical across rows (three experiments) and the third and fourth columns also identical across rows (three additional experiments). Shown in the second and fourth columns are same-size (within-image) circles centered on the different T4 plaques. Phage T7 plaques are as labeled and are too large to be viewed in their entirety within a single field of view, as were photographed through the ocular lens of a dissecting microscope. Of interest is the intersection between the T7 plaque edges and these circles, which are suggestive of less visually obvious boundaries to the breadth of the zones of infection of T4 wild-type plaques. Note, though, that the T7 plaques invade the T4 wild-type plaques more deeply than do T4 *r*48 plaques (Figure 4), that is, as heading towards the wild-type T4 plaque centers. This deeper invasion I interpret as being due to faster phage T7 plaque growth than that of phage T4, with T7 plaques thereby reaching the T4 wild-type-infected bacteria sooner, and therefore before the T4 plaques had grown fully in size. By contrast, we predict that wild-type T4 and *r* mutant plaques, as shown in Figure 4, would display more similar rates of ‘plaque’ growth. The conclusion proposed from this experiment is that the zone of infection for phages T4 *r*48 and wild-type are similar in diameters. This figure is based on experiments equivalent to one originally published in Abedon [80], which is reprinted in the figure in part (lower-right), with permission from Nova Science Publishers, Inc. Plaques were generated by Cameron Thomas-Abedon and most of these examples are otherwise unpublished. No chloroform was used in these experiments.

**Figure 6 viruses-11-00951-f006:**
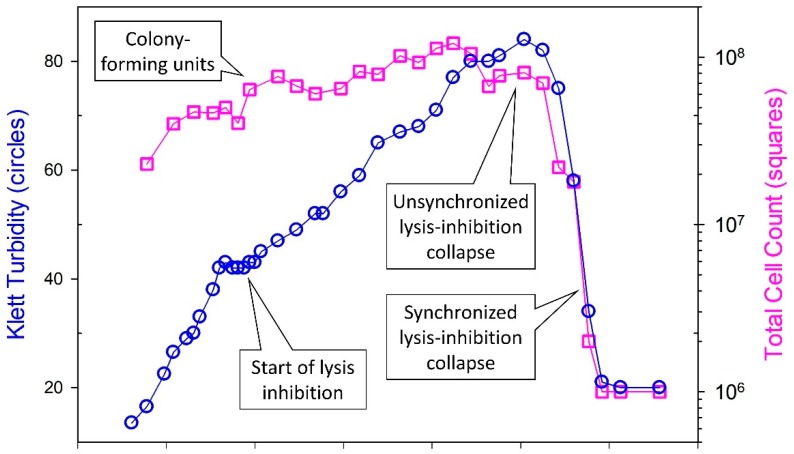
Impact of lysis inhibited T4-infected bacteria on free T4 phages. Lysis inhibition-expressing, double amber mutant T4 phages (*42amNG205* × *43am4306*) were added at time zero to *E. coli* CR63 bacteria (an amber-suppressing strain) with a multiplicity of 0.2. The resulting phage infections were then followed by both turbidity (blue circles, top panel) and total cell count per ml (pink squares, top and bottom panels). At various intervals, aliquots of culture were removed to which wild-type, free T4 phages (T4D) were added at low multiplicity (<<1). These now-separate cultures were then followed in terms of titers using non-amber suppressing *E. coli* K12 wild-type indicator bacteria (stars, bottom panel; titer data normalized to a staring density of 100 = 10^2^ plaque-forming units/mL; T4D is a wild-type laboratory strain of phage T4). *E. coli* K12, unlike *E. coli* CR63, is non-permissive to the double amber mutant phages and therefore upon plating only the added T4D phages, particularly as free phages, are able to form plaques. Thus, the generated starred curves are a series of added T4D-only free-phage adsorption curves [55]. Indicated are declines in T4D titers over 7.5-min spans, for a total of 11 such curves (all shown, lower panel). Note that rates of decline in titers are drastically reduced once culture lysis has occurred, i.e., after approximately 225 min; see especially curve numbers 7 through 11 for indication of this reduced rate of decline. Note that the increase in total cell counts observed between roughly 50 and 150 min is not seen consistently in all experiments [18]. An equivalent experiment is published in Abedon [6] and both were performed by David Brennan working under my supervision.

**Figure 7 viruses-11-00951-f007:**
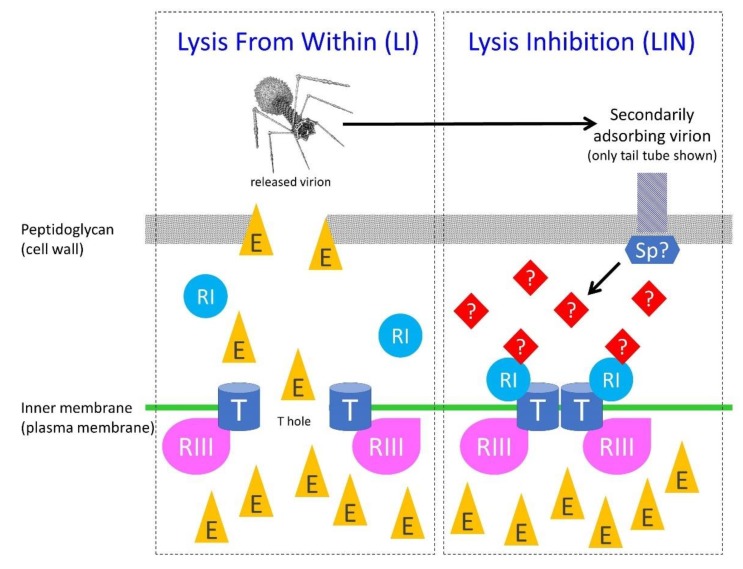
T-hole formation and its inhibition: lysis from within (LI; left) versus lysis inhibition (LIN; right). LI is caused by plasma membrane disruption as mediated by the phage T-holin protein and resulting T holes. With secondary adsorption, T-hole formation can be blocked. This occurs due to the action of the periplasmic phage RI protein, an antiholin. The RI protein presumably is functional as an antiholin particularly as complexed with a substance supplied by the secondarily adsorbing phage, e.g., such as periplasmic DNA (indicated as a question mark-carrying red diamond). The phage Sp protein—as supplied intracellularly by the primary phage—may or may not be required for release of this LIN-inducing substance. The phage Imm protein (not shown, but also as supplied by the primary phage intracellularly) might contribute as well to this LIN-inducing substance’s release into the periplasm of the secondarily adsorbed bacterium. The phage RIII protein, also an antiholin, inhibits T-hole formation from the cytoplasm, though this inhibition occurs independent of secondary adsorption. Phage protein E is the phage endolysin, commonly described for phage T4 as a lysozyme. With LI, T-hole formation allows E lysozyme access to the cell wall of the phage-infected bacterium, resulting in cell wall degradation and resulting bacterial lysis. LI thereby gives rise to release of intracellularly matured phage virions, and these now free virions are then available to infect phage uninfected bacteria as primary phages or available to adsorb already phage-infected bacteria as secondary phages. The latter for T4 phages induces LIN, but also likely contributes to a synchronized LIN collapse (as well as, though not shown in the figure, lysis from without, LO). All schematically presented proteins are as expressed by primary phages, i.e., E, T, RI, and RIII as well as the already so-noted Sp and not shown Imm.

**Figure 8 viruses-11-00951-f008:**
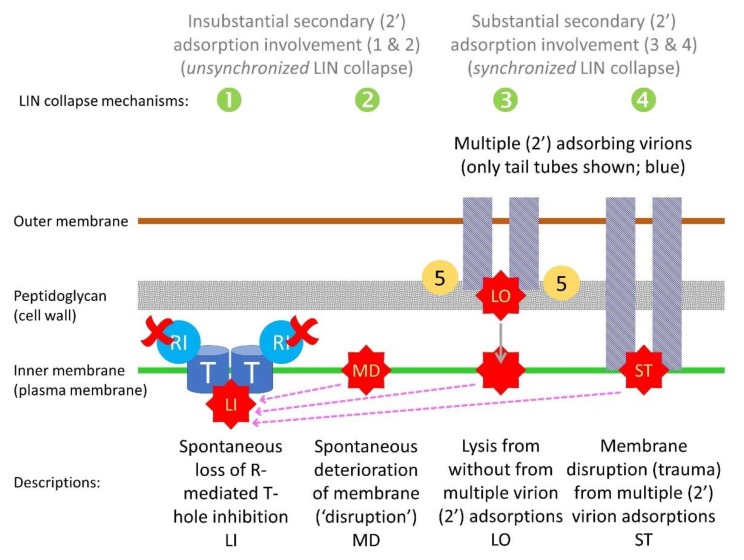
Possible mechanisms leading to lysis-inhibition collapse. ❶ Loss of R-mediated inhibition of T-hole formation is indicated with the ***⨯*** marks, resulting in lysis from within (LI). Note that protein RIII is not illustrated in the figure for the sake of reducing clutter. ❷ Membrane deterioration (MD) results in a spontaneous loss of plasma membrane function as a chemiosmotic barrier, likely leading to LI (top dashed arrow as leading to mechanism 1). ❸ Lysis from without as caused by the gene *5* lysozyme product, which is associated with T4 virions, can result from multiple virion secondary (2′) adsorptions. The associated cell-wall disruption gives rise to plasma membrane disruption (as indicated by the vertical, gray arrow), thereby presumably leading LINed bacterial infections also to LI (middle dashed arrow). ❹ Perhaps independent of LO is possible membrane disruption also caused by excessive phage secondary adsorptions, giving rise to what can be described as a secondary traumatization (ST), with disruption of the plasma membrane chemiosmotic barrier again potentially resulting in LI (lower dashed arrow). The R-mediated inhibition of LI, as would be overcome for mechanism 1, is discussed in Section 5.2. Section 5.3.2 discusses evidence against the significance of mechanism 2 to LIN collapse. Mechanism 3 likely is inhibited to some degree by gene *sp* expression by primary phages (Section 5.4, but not illustrated in the figure). Mechanism 4 possibly is inhibited to some degree by gene *imm* expression (also Section 5.4, and also not illustrated in the figure).

**Figure 9 viruses-11-00951-f009:**
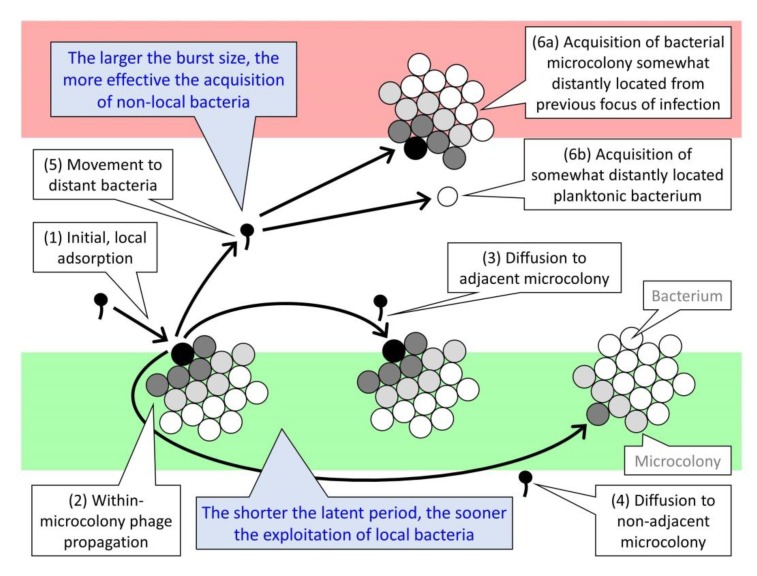
A model of the ecology of biofilm exploitation by bacteriophages, here as differentiated in terms of more localized versus more distant bacterial targets. The green-colored stripe along the bottom of the figure represents the focus biofilm, with a focus of phage infection as initiated, e.g., by a single phage virion particle (**1**). This focus of infection is associated with various bacterial microcolonies, in this case consisting of bacteria which are similarly susceptible to the phage in question. The red-colored stripe across the top represents a somewhat spatially distant biofilm, i.e., as could be centimeters or more separated from the lower biofilm. In this second biofilm, only a single associated microcolony is illustrated. The more virions produced in association with the lower biofilm—especially given virion dilution into the environment and inevitable virion decay over time—then the greater the likelihood that the upper biofilm will be discovered by phages released from the lower biofilm (**6a**), or instead that planktonic bacteria will be encountered and then adsorbed (**6b**). Notwithstanding that fewer virions would be produced, given shorter phage latent periods, then faster local exploration of biofilms should be possible (Section 6.1). This is illustrated within the lower biofilm (**2**, **3**, and **4**). In addition, with shorter phage latent periods, then the sooner that phage virions may leave this biofilm in search of new biofilms to exploit (**5**), albeit with fewer phages released per phage-infected bacterium. Lysis inhibition, by being an inducible latent-period extension, thereby could allow for both strategies: (i) shorter latent periods during local exploration for new bacteria to infect as well as sooner virion spread out of biofilms towards more distant bacteria, but also (ii) greater virion production towards increasing the potential to discover those more distant bacteria. See Section 4.1.4 for a description of an analogous scenario during wild-type phage T4 plaque growth. The figure is derived and modified from those found in Abedon [80,121,122].

**Table 1 viruses-11-00951-t001:** Abbreviations used (see Table 2 for fuller definitions).

Abbreviation	Stands for…
5 (*5*)	Gene product *5* (phage T4 protein and associated gene)
AAPI	*A*utoinducer-*A*ssociated *P*rophage *I*nduction
AS	*A*rbitrium *S*ystem
E (*e*)	*E*ndolysin (phage T4 protein and associated gene)
HMLD	*H*igh-*M*ultiplicity *L*ysogeny *D*ecision
Imm (*imm*)	*Imm*unity (phage T4 protein and associated gene)
LI	*L*ysis from with*I*n
LIN	*L*ysis *IN*hibition
LINed	*L*ysis *IN*hibit*ed*
LIN collapse	*L*ysis-*IN*hibition *collapse*
LO	*L*ysis from with*O*ut
MD	*M*embrane *D*eterioration
R (*r*)	*R*apid lysis
SA	*S*econdary *A*dsorption
Sp (*sp*)	*Sp*ackle (phage T4 protein and associated gene)
ST	*S*econdary *T*raumatization
T (*t*)	*T*ithonus (phage T4 protein and associated gene)

**Table 2 viruses-11-00951-t002:** Terms and concepts used.

Term or Abbreviation	Meaning	Overview or Discussion
5 (*5*)	Gene product 5; *5* is the underlying phage T4 gene	Protein making up the phage T4 virion tail tube tip, which is a lysis from without (LO), cell-wall digesting lysozyme
Arbitrium System (AS)	Phage-encoded, autoinducer-mediated, lysis delay	As achieved by temperate phages, resulting in lysogenic rather than lytic cycles; an example of phage-associated intercellularly mediated communication
Autoinducer	Quorum-sensing signaling molecule	Generally, a bacterium-produced molecule but also as encoded by certain temperate phages, re: arbitrium systems (ASs)
Autoinducer-associated prophage induction (AAPI)	Quorum-sensing autoinducer-mediated lysis acceleration that is associated with prophage induction	As has been found in association with *V. cholerae*; an example of phage-associated intercellularly mediated communication
Coinfection	Infection of a cell by more than one phage	A consequence of simultaneous or secondary infection; generally, lysis inhibition (LIN) is not explicitly a coinfection-associated phenomenon
E (*e*)	*E*ndolysin	The phage T4 lysis from within (LI), cell-wall digesting lysozyme protein, as encoded by gene *e*
Free phage (free virion)	A post-release mature phage virion, i.e., as not still found within its parental phage infection	Though phage virions can be fully mature prior to release, it is only free phages which represent a bacterium adsorption-capable phage state
Focus of infection	Localized, potentially plaque-like region of phage population growth found in association with a bacterial biofilm	The phage potential to discover new biofilms to exploit likely is function of the number of virions produced, and then disseminated, per individual focus of infection
High-multiplicity lysogeny decisions (HMLDs)	Coinfection-associated lysis delay by a temperate phage infection	Lysis delay is achieved with HMLDs by biasing lytic-lysogeny decisions towards lysogeny; an example of phage-associated intercellularly mediated communication
Homoimmune	Possessing the same temperate phage immunity type	Superinfection immunity is imposed upon homoimmune secondarily infecting phages; note that neither homoimmunity nor superinfection immunity are associated with phage T4 gene *imm*
Imm (*imm*)	*Imm*unity	Phage T4 protein, as encoded by gene *imm*, that is associated with phage expression of superinfection exclusion, resistance to lysis from without (LO; though is a lesser component of resistance to LO than protein Sp), resistance to secondary traumatization (ST), and also resistance to a premature lysis-inhibition (LIN) collapse
Induction (prophage)	Conversion of a latent (lysogenic) infection into a productive infection	Canonically, i.e., as with phage lambda, this prophage induction is associated with bacterial-host DNA damage and resulting SOS response
Lysis acceleration	Occurrence of sooner phage-induced bacterial lysis	As associated with (i) temperate-phage display of lytic rather than lysogenic cycles, (ii) premature termination of lytic cycles (re: premature LIN collapse or synchronized LIN collapse), or (iii) prophage induction during lysogenic cycles
Lysis delay	Later lysis; longer phage infection (latent) period, including as achieved by lysogenic cycles	As associated with (i) delayed termination of lytic cycles such as seen with lysis inhibition (LIN) or unsynchronized LIN collapse, (ii) decisions to display lysogenic cycles during lytic-lysogeny decisions, or (iii) ongoing display of lysogenic cycles rather than prophage induction
Lysis from within (LI)	Phage-induced bacterial lysis occurring at the end of phage lytic infections as stimulated intracellularly	With T4 phages, LI is associated, at a minimum, with genes *e* (endolysin) and *t* (holin); contrast LI with lysis from without (LO); see also unsynchronized lysis-inhibition (LIN) collapse; LI is a possible mechanism (mechanism 1) underlying at least certain aspects of lysis-inhibition (LIN) collapse
Lysis from without (LO)	Phage-induced bacterial lysis that is dependent especially on multiple phage adsorptions, thus as stimulated extracellularly	LO technically is not dependent on phage infection of a bacterium; contrast lysis from within (LI); LO is a possible mechanism (mechanism 3) underlying at least certain aspects of lysis-inhibition (LIN) collapse, particularly synchronized lysis-inhibition (LIN) collapse
Lysis inhibition (LIN)	Multiple virion-adsorption- (secondary adsorption-) associated, inducible lytic cycle lysis delay	Lysis inhibition results in an extended primary infection lytic cycle and resulting increase in infection burst size; LIN is an example of phage-associated intercellularly mediated communication
Lysis-inhibition (LIN) collapse	Lysis of lysis-inhibited phage infections (see possible mechanistic underpinnings, 1 through 4, immediately below)	LIN collapse does not imply substantial synchronization of lysis across a LINed culture nor necessarily a lack of lysis synchronization (unsynchronized LIN collapse); synchronized LIN collapse is a possible example of phage-associated intercellularly mediated communication
Lysis-inhibition collapse, proposed mechanism 1	As associated especially with lysis from within (LI)	Reversal of R-protein associated inhibition of T-hole formation
Lysis-inhibition collapse, proposed mechanism 2	As associated especially with membrane deterioration (MD)	Spontaneous loss of plasma membrane stability as potentially leading to lysis from within (LI)
Lysis-inhibition collapse, proposed mechanism 3	As associated especially with lysis from without (LO)	Secondary adsorption-associated loss of cell-wall stability as potentially leading to LI
Lysis-inhibition collapse, proposed mechanism 4	As associated especially with secondary traumatization (ST)	Secondary adsorption-associated loss of plasma membrane stability as potentially leading to LI
Lysogenic cycle	Non virion-productive, but otherwise phage-genome replicative temperate phage latent infection	During lysogenic cycles phages exist as prophages and do not produce virion progeny; both the occurrence and extensions of lysogenic cycles constitute lysis delays
Lytic-lysogeny decision	Choice that must be made at the start of temperate phage infections	Depending on conditions, this choice may be biased either towards or away from display of lysogenic cycles (as representing delayed lysis), though lytic cycles (representing accelerated lysis) tend to be the default decisions
Lytic cycle	Productive phage infection which ends in infection lysis	During lytic cycles, phages are committed to producing phage virions and, if successful, then infected host bacteria do not survive; extensions of the duration of lytic cycles represent lysis delays, whereas earlier lysis represents lysis acceleration
Membrane deterioration (MD)	Nonspecific spontaneous deterioration of plasma membranes as potentially inducing lysis from within (LI)	A mechanism (mechanism 2) potentially underlying certain aspects of lysis-inhibition (LIN) collapse, particularly unsynchronized LIN collapse
Premature lysis-inhibition (LIN) collapse	Earlier than expected lysis of LINed culture (accelerated lysis)	Such as might be caused by excessive lysis from without- (LO-) like secondary adsorption-associated damage to otherwise lysis-inhibited (LINed) bacteria infected with *sp* or *imm* mutant phages
Primary infection or phage	Infection of a cell by only a single phage or referring to the first phage to reach and infect a cell	Primary infections may display superinfection exclusion or superinfection immunity against secondarily adsorbing phages; it is primary infections that both encode and display lysis inhibition (LIN)
Productive infection	Phage infection in which virion progeny are both produced and released	Both rapid lysis lytic cycles and lysis-inhibited (LINed) infections are productive infections, while lysogenic cycles by definition are not virion productive
Prophage	Temperate phage genome as observed during lysogenic cycles	Prophages are generated following lytic-lysogeny decisions (given a lysogeny decision) and are lost given prophage induction
Prophage induction	Productive termination of a lysogenic cycle	This can be viewed as lysis acceleration as observed in a context of a temperate phage lysogenic cycle; see also induction (prophage)
Rapid lysis	Constitutively non-lysis inhibited latent period	The phenotype associated with a genetic inability to display lysis inhibition (LIN) but an ability to still display lytic cycles is described as rapid lysis
R (*r*)	*R*apid lysis, as mutated in rapid-lysis (*r*) phage mutants	Products of rapid-lysis (*r*) genes include the RI, RIIA, RIIB, and RIII proteins, as required for lysis-inhibition (LIN) expression
Resistance to lysis from without	Phage-encoded minimization of cell wall damage caused by phage secondary adsorption	In T4 phages this resistance is associated with gene *imm* and especially with gene *sp*
Restrict	The killing of a phage upon its adsorption or infection of a bacterium	Phage restriction is mediated by superinfection exclusion as well as by superinfection immunity
Secondary adsorption (SA)	Attachment of a virion to an already phage-infected cell	Typically described as superinfection, but secondary adsorption as a term is used here instead to avoid implying that secondary infection necessarily always occurs following secondary adsorption
Secondary infection	Infection by a virion of an already phage-infected cell	Infection here is defined as successful phage genome entry into an adsorbed cell’s cytoplasm; superinfection exclusion specifically blocks the initiation of secondary infections by secondarily adsorbing virions
Secondary traumatization (ST)	Death of T4-infected bacteria due to excessive secondary adsorption but not as due to lysis from without	Not strictly associated with phage-infection lysis and thereby not strictly equivalent to lysis from without (LO); ST is a possible mechanism (mechanism 4) underlying at least certain aspects of lysis-inhibition (LIN) collapse, particularly synchronized lysis-inhibition (LIN) collapse
Sp (*sp*)	*Sp*ackle	Phage T4 protein, as encoded by gene *sp*, associated with expressing both superinfection exclusion and resistance to lysis from without (LO); likely also associated with expression of a resistance to premature lysis-inhibition (LIN) collapse
Strictly lytic	Description of a lytic phage which is unable to display lysogenic cycles	Also known as obligately lytic, professionally lytic, or virulent; contrast with temperate phage
Superinfection	Virion infection of an already phage-infected cell	Often in the literature superinfection is not rigorously distinguished from simply secondary adsorption
Superinfection exclusion	Block on phage infection, but not on phage adsorption	Superinfection exclusion is imposed post virion attachment but prior to successful phage DNA translocation into the bacterial cytoplasm; it is a form of phage restriction; contrast with superinfection immunity
Superinfection immunity	Block on phage infection which occurs post successful phage DNA translocation into the bacterial cytoplasm	Superinfection immunity particularly is as associated with superinfection, by temperate phages, of homoimmune phage lysogens, and is a form of phage restriction; contrast with superinfection exclusion
Synchronized lysis-inhibition (LIN) collapse	Multiple virion secondary adsorption-associated, coerced, accelerated LIN collapse	As resulting in faster-than-may-otherwise-be-expected lysis of a lysis-inhibited (LINed) culture once LIN collapse has begun; see lysis-inhibition (LIN) collapse proposed mechanisms 3 and 4; contrast with unsynchronized lysis-inhibition (LIN) collapse
T (*t*)	*T*ithonus (of Greek mythology, who was to become immortally old, as so too, arguably, do the infections of never-lysing gene *t* knock-out mutants)	The phage T4 holin protein as encoded by gene *t* is responsible for controlling the timing of infection lysis as well as allowing otherwise cytoplasmic E protein to access the bacterial cell wall from within, resulting in lysis from within (LI)
T-hole	*T* protein-associated plasma membrane *hole*	Product of holin activation, resulting in a hole in an infected bacterium’s plasma membrane through which otherwise cytoplasmic E-lysozyme protein can diffuse
T-holin	*T* protein, which is a *holin*	This construct is used here simply to clarify the function of T protein
Temperate phage	Lysogenic cycle-capable bacteriophage	Lytic temperate phages can display both lytic cycles and lysogenic cycles, but not both simultaneously; contrast with phages that are strictly lytic
Unsynchronized lysis-inhibition (LIN) collapse	LIN collapse that is *not* directly associated with a multiple virion secondary adsorption-coerced lysis acceleration	See lysis-inhibition (LIN) collapse proposed mechanisms 1 and 2; contrast with synchronized lysis-inhibition (LIN) collapse
Zone of infection	Area associated with a phage plaque that contains either phage virions or phage-infected bacteria but is not necessarily visible to the eye	The zones of infection of wild-type phage T4 plaques may be somewhat larger than the visible clearings associated with these plaques

**Table 3 viruses-11-00951-t003:** Summary of Golec et al. [96] results.

Specific Experimental Results	Phage	Over-Expression from a Plasmid of				Effect
		gene *rI*	gene *rIII*	gene *rI.1*	gene *rI.-1*	
A	WT			✓		Slightly delayed LIN collapse
A	WT				✓	Slightly delayed LIN collapse
B	WT		✓			Greatly delayed LIN collapse
C	WT	✓				Little impact
D	WT	✓	✓			Slightly delayed LIN collapse
E	WT	✓		✓		Somewhat delayed LIN collapse
F	*rIII*		✓			Greatly delayed LIN collapse
G	*rIII*		✓	✓		Slightly delayed LIN collapse
H	*rIII*		✓		✓	Less than greatly delayed LIN collapse
I	none			✓		Toxicity to bacteria
J	none		✓	✓		Absence of toxicity to bacteria
K	none	✓	✓	✓		Toxicity to bacteria
L	none		✓			Slowed bacterial growth
M	none		✓		✓	Absence of toxicity to bacteria
M	none		✓	✓		Absence of toxicity to bacteria
^1^	none	✓				Absence of toxicity to bacteria

^1^ Not discussed in main text.

**Table 4 viruses-11-00951-t004:** Hypotheses Regarding the Mechanism(s) of LIN Collapse and its synchronization.

Associated with…	Hypothesized LIN Collapse Mechanisms			
	1 (LI)	2 (MD)	3 (LO)	4 (ST)
LIN collapse	Yes	Yes	Yes	Yes
Synchronized LIN collapse	No	No	Yes	Yes
Unsynchronized LIN collapse	Yes	Yes	No	No
E lysozyme	Yes	Yes	No ^1^	Yes
T holin	Yes	Yes	No ^1^	Yes
RI antiholin inactivation	Yes	No	No	No
Membrane deterioration (MD) ^2^	No	Yes	No	Yes
Secondary adsorption (SA) ^3^	No	No	Yes	Yes
Lysis from within (LI)	Yes	Yes ^4^	No ^1^	Yes ^5^
Lysis from without (LO)	No	No	Yes	No
Secondary traumatization (ST) ^6^	No	No	No	Yes ^7^
Well-established mechanism ^8^	Yes	No	Yes	No

^1^ Given the presence of proteins T and E in LINed bacteria, it is possible that these proteins could contribute to bacterial lysis once LO has begun in a given bacterium, i.e., with LO augmented by LI, but LI nevertheless is not emphasized as a contributor to LIN collapse in this column (3). ^2^ This is membrane deterioration (MD) occurring prior to rather than following T-hole formation, i.e., as equivalent to Young’s [20] “nonspecific deterioration of the membrane”. An expectation is that substantial breaching of the plasma membrane such as due to MD would result in a triggering of T-hole formation and thereby a subsequent approximation of normal LI. ^3^ A “Yes” meaning as caused by secondary adsorption (SA), rather than occurring independent of SA, and also occurring in relatively close temporal proximity to SA, e.g., not tens or more minutes subsequent to SA. ^4^ LI as stemming from membrane deterioration (MD) but as possibly associated with SA, though as occurring in a somewhat delayed manner, e.g., tens or more minutes subsequent to SA. ^5^ LI as stemming from MD that has been immediately caused by SA, rather than a delayed consequence of much earlier (tens of minutes) SA. ^6^ Secondary traumatization (ST) technically is infection death due to high multiplicity phage SA that cannot be explained by LO of the bacterium [102]. As this effect is seen especially with phage T4 imm mutant primary phages, which may be defective in resistance to plasma-membrane damage caused by SA (see Section 5.4), ST could be a consequence of such SA-caused plasma-membrane damage. ^7^ Speculation. ^8^ Other than in terms of LIN collapse.

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
