# Peer review of "Look Who’s Talking: T-Even Phage Lysis Inhibition, the Granddaddy of Virus-Virus Intercellular Communication Research"

_viruses, 2019, doi:10.3390/v11100951_

Round 1

Reviewer 1 Report

The manuscript of Stephen Abedon is a review on phage lysis inhibition. Overall, the effort of putting this kind of virus-virus communication into the bigger picture - especially in times of novel findings on arbitrium systems and autoinducer controlled lysis-lysogeny decisions - is definitely worthwhile. The review provides an historical overview on the topic of lysis inhibition, followed by a description of phenotypes (showcasing examples from the own lab), followed by a section on the mechanistic basis of LIN and concluding with sections on the evolutionary/ecological role of LIN. The review is principally well-structured and nicely written, but some sections would maybe benefit from a more concise presentation (my personal opinion). Overall, I am supportive of publication after consideration of the following comments:

At some places in the text (e.g. l. 1081), the reader gets the impression that the author does not like the “fan-fare” about the “rediscovered” virus-virus communication by, e.g. the arbitrium systems or the autoinducer controlled lysis-lysogeny decision. However, research on these systems is at very high level and provided unprecedented mechanistic insights – which is ultimately a benefit for all of us working in this field. Therefore, I would reconsider the wording here.

My major concern is actually regarding the figures. These could be more informative and could (where possible) include more mechanistic details. Especially figure 2 is not presented in a catchy way and does not add to the understanding (in comparison to just reading the text).

Figure 1, appropriate references should be added for the different mechanisms.

When discussing the plaque types, the author is frequently referring to the effect of chloroform treatment. If I understood this correctly, the author is suggesting that the observed phenotype is the result of an even wider spread of infection that it can be seen simply from the plaque size? This could be nicely visualized by a simple scheme going along with the plaque pictures.

Overall, this section 4 on plaque phenotypes could be shortened - in my opinion.

Section 5: “Mechanisms” – here I would strongly suggest to incorporate a meaningful figure comparing the different mechanisms. Figure 8 is kept very simple, but it doesn’t really help me to understand what’s the underlying mechanism (LIN & LIN collapse). The same holds also for Table 2. It doesn’t really help me to get a clearer picture of the (molecular) mechanisms. → I would suggest to prepare a figure showing what is known regarding the molecular mechanisms underlying LIN as described in 5.2.

Figure 8, the author only cites his own study from 1992. Here, also reference for the reports on the describe mechanisms should be added.

Author Response

I WOULD LIKE TO THANK THE REVIEWER FOR THEIR NUMEROUS HELPFUL COMMENTS. I HAVE DONE MY BEST TO INCORPORATE THESE INTO THE MANUSCRIPT.

NOTE, HOWEVER, THAT IN DOING SO, AS WELL AS IN ADDRESSING THE COMMENTS PRESENTED BY OTHER REVIEWERS, THE MANUSCRIPT IS NOW SOMEWHAT LONGER OVERALL THAN IT HAD PREVIOUSLY BEEN (IN PARTICULAR, TWO SUBSECTIONS, ONE TABLE, AND TWO FIGURES HAVE BEEN ADDED, THE LATTER AS REQUESTED BY THIS REVIEWER, THOUGH THREE FIGURES HAVE ALSO BEEN REMOVED). I NONETHELESS HAVE GONE THROUGH THE MANUSCRIPT AND DELETED VARIOUS TEXT THAT PERHAPS WERE SUPERFLUOUS, MOST NOTABLY REMOVING WORDS FROM THE SUBSECTIONS ON PLAQUES (AS EXPLICITLY REQUESTED), REMOVING THE THREE FIGURES, AND REMOVING THE SECTION ON PRISONER’S DILEMMA’S, THE LATTER ALSO AS REQUESTED BY A DIFFERENT REVIEWER, AND .

The manuscript of Stephen Abedon is a review on phage lysis inhibition. Overall, the effort of putting this kind of virus-virus communication into the bigger picture - especially in times of novel findings on arbitrium systems and autoinducer controlled lysis-lysogeny decisions - is definitely worthwhile. The review provides an historical overview on the topic of lysis inhibition, followed by a description of phenotypes (showcasing examples from the own lab), followed by a section on the mechanistic basis of LIN and concluding with sections on the evolutionary/ecological role of LIN. The review is principally well-structured and nicely written,

Comment 1.1:

but some sections would maybe benefit from a more concise presentation (my personal opinion). Overall, I am supportive of publication after consideration of the following comments:

Reply 1.1:

I HAVE DONE MY BEST TO TRY TO MAKE THINGS MORE CONCISE, THOUGH HAVE ALSO ADDED WORDS AND OTHERWISE REJIGGERED SECTIONS TOWARDS IMPROVED UNDERSTANDABILITY AND FLOW, AS WELL AS IN RESPONSE TO VARIOUS REVIEWERS’ COMMENTS, AND THEN IN FURTHER RESPONSE TO THE LATTER CHANGES. OVERALL, AS A CONSEQUENCE, THE MANUSCRIPT HAS BEEN SUBSTANTIALLY REWRITTEN, I AM HOPEFUL FOR THE BETTER.

Comment 1.2:

At some places in the text (e.g. l. 1081), the reader gets the impression that the author does not like the “fan-fare” about the “rediscovered” virus-virus communication by, e.g. the arbitrium systems or the autoinducer controlled lysis-lysogeny decision. However, research on these systems is at very high level and provided unprecedented mechanistic insights – which is ultimately a benefit for all of us working in this field. Therefore, I would reconsider the wording here.

Reply 1.2:

REPLY: ACTUALLY, HONESTLY, I THINK ITS GREAT THAT VIRUS-VIRUS COMMUNICATION HAS BEEN REDISCOVERED AND IS FINALLY BEING TAKEN SERIOUSLY AS AN ECOLOGICAL OR EVOLUTIONARY ECOLOGICAL PHENOMENON. INDEED, FOR ME IT MEANS THAT SOMETHING THAT I’VE BEEN WORKING ON AND THINKING ABOUT FOR 30 YEARS NOW MIGHT ‘SUDDENLY’ BE MORE RELEVANT (I.E., LYSIS INHIBITION AND LIN COLLAPSE). NEVERTHELESS, TO ADDRESS THIS PERCEIVED SLIGHT FROM MY PROSE, I’VE CHANGED “, TO MUCH FAN-FARE” INSTEAD TO “IN THE GUISE OF THE NEWLY DISCOVERED PHAGE ARBITRIUM SYSTEM”, AND “REDISCOVERED” TO “DESCRIBED”.

Comment 1.3:

My major concern is actually regarding the figures. These could be more informative and could (where possible) include more mechanistic details. Especially figure 2 is not presented in a catchy way and does not add to the understanding (in comparison to just reading the text).

Reply 1.3:

I AM IN AGREEMENT THAT FIGURE 2 IS JUST A RESTATEMENT OF WHAT IS FOUND IN THE TEXT. I’M NOT SURE THAT THERE IS ROOM IN THAT FIGURE FOR ADDITIONAL MECHANISTIC DETAIL SO I HAVE CUT IT FROM THE MANUSCRIPT, PARTICULARLY GIVEN THE SUGGESTION BELOW THAT A SOMEWHAT PURELY MECHANISTIC FIGURE BE ADDED LATER IN THE MANUSCRIPT.

IN RESPONSE TO THE SECOND REVIEWER’S COMMENTS, I HAVE ALSO BEEFED UP THE LEGEND TO WHAT IS NOW FIGURE 6 AND ALSO HAVE ADDED LABELING TO THE FIGURE (THE FIGURE IS THE ONE SHOWING THE FREE PHAGE ADSORPTION CURVE IN ASSOCIATION WITH AN LINed LYSIS PROFILE). IN RESPONSE TO THIS REVIEWER’S FURTHER COMMENTS (BELOW), I HAVE ADDED A MECHANISTIC SCHEMATIC OF INHIBITION OF T-HOLE FORMATION (NOW FIGURE 7) AND DELETED FORMER FIGURE 10 (UTILITY OF SYNCHRONIZED LINE COLLAPSE), REPLACING IT WITH ANOTHER MORE MECHANISTIC SCHEMATIC (NOW FIGURE 8).

Comment 1.4:

Figure 1, appropriate references should be added for the different mechanisms.

Reply 1.4:

ALL REFERENCES TO THE VARIOUS PHENOMENON THAT I AM AWARE OF HAVE BEEN ADDED TO THE FIGURE LEGEND, EXCEPT FOR NUMEROUS EARLY REFERENCES TO PHAGE r MUTANTS, THOUGH I CITE A REVIEW (MINE FROM 1994) THAT DIRECTS READERS TO AT LEAST SOME OF THOSE ADDITIONAL REFERENCES.

Comment 1.5:

When discussing the plaque types, the author is frequently referring to the effect of chloroform treatment. If I understood this correctly, the author is suggesting that the observed phenotype is the result of an even wider spread of infection that it can be seen simply from the plaque size? This could be nicely visualized by a simple scheme going along with the plaque pictures.

Reply 1.5:

I HAVE ADDED A SCHEMATIC, AS WELL AS DISCUSSION TO THE FIGURE (3a), TO IMPROVE READER APPRECIATION OF WHAT THESE EXPERIMENTS MAY BE INDICATING.

Comment 1.6:

Overall, this section 4 on plaque phenotypes could be shortened - in my opinion.

Reply 1.6:

I HAVE CUT ABOUT 500 WORDS FROM THIS SECTION’S MAIN TEXT WITH ABOUT 1100 WORDS NOW REMAINING.

Comment 1.7:

Section 5: “Mechanisms” – here I would strongly suggest to incorporate a meaningful figure comparing the different mechanisms. Figure 8 is kept very simple, but it doesn’t really help me to understand what’s the underlying mechanism (LIN & LIN collapse). The same holds also for Table 2. It doesn’t really help me to get a clearer picture of the (molecular) mechanisms. → I would suggest to prepare a figure showing what is known regarding the molecular mechanisms underlying LIN as described in 5.2.

Reply 1.7:

I HAVE ADDED TWO NEW FIGURES, “T-HOLE FORMATION AND ITS INHIBITION: LYSIS FROM WITHIN VERSUS LYSIS INHIBITION” AND “POSSIBLE MECHANISMS LEADING TO LYSIS-INHIBITION COLLAPSE”. YES, I AGREE, THESE SHOULD BE USEFUL TO THE READER. THE LATTER HAS REPLACED WHAT HAD BEEN FIGURE 10.

Comment 1.8:

Figure 8, the author only cites his own study from 1992. Here, also reference for the reports on the describe mechanisms should be added.

Reply 1.8:

I HAVE REPLACED THIS FIGURE AS INDICATED ABOVE. NO REFERENCES ARE FOUND IN NEW VERSION OF THE FIGURE. THE READER INSTEAD IS DIRECTED TO APPROPRIATE SECTIONS OF THE MAIN TEXT.

Reviewer 2 Report

This paper presents a thorough background on lysis inhibition in T4 bacteriophage (as phenotype, molecular mechanism, and adaptation), while connecting this system to more recent developments in phage communication. I suspect it would be useful as a review of the system, especially for those newly entering into the field. However, it does suffer from some imprecision with regards to the nature of communication, along with often unclear and overly verbose writing.

Overall, organizing the paper so it is clearer how the ideas will be important to understanding LIN and LIN collapse as communication would be helpful. I would also go through carefully and make edits to improve the clarity of your writing.

Comments below are organized by line. Bolded comments are more general and important ones.

31: A brief introduction to what communication IS, or at least how you are defining it, would provide a useful introduction, especially since your arguments later in the article depend crucially on said definition.

84: This is not a good approach, and lends itself to being misleading. Don’t “use ‘T4’ as a stand-in for T-even phages”. Instead use T4, and it you want to extend out to other T-even phages say so. As far as I can tell you primarily talked about T4 in any case. But because of this statement I have no way of knowing if at some point you wrote something about some different T-even phage and called it T4. You could instead say that T4 is fairly similar to the other T-even phages and thus likely to be representative.

103: of->a

123: You do not need this many abbreviations, and they make your paper harder to follow. LIN is useful, but MD for “membrane deterioration” for example is superfluous. You only use it 4 times without writing out the whole thing. There are a number of other superfluous abbreviations. If this seems like a small point, I think it reflects in general that you frequently introduce jargon, and in fact new jargon that is not widespread in the literature, in a way that makes your writing more difficult to understand but does not add much meaning. This may also in some cases make widely understood concepts seem more recondite and original than they are.

The existence of the table, on the other hand, is helpful, especially to any readers who are interested in only part of the paper.

167: This figure doesn’t demonstrate this! There’s no mutant present.

194: This is one example of something that I see repeatedly in this paper – a tendency to insert yourself into the center of the story. This is unnecessary and distracting. Of course it is your opinion – you are writing the article. There are ways to write the same thing without inserting yourself, for example you might say that the discovery “may be” one of the more exciting ones if you wish to emphasize potential differences of opinion. Please search for “my” and “I” in this paper and consider in each case whether it is useful. Note that this isn’t about active voice, which is fine as far as I am concerned. It’s about unnecessary self-insertion that adds little to the story. See especially line 922 and 935.

241: Again, the figure doesn’t show this.

244: This section was interesting, convincing and informative, at least to someone like me who had not read reference 62. However, it is not entirely clear to me how it fits into the paper’s framework of LIN and LIN collapse as communication. I think there might be a way to tie it in.

414 and 426: Don’t order your reader around...

471: I spent a while thinking about what these mutations were supposed to do before I figured out they were simply to create nonpermissive cells. A brief description of the rationale for the experiment would have made this easier.

536: I do not know what it means to tell the story from their perspective. Do they disagree with you?

603: I think these different mechanisms, or rather discussing them all and the differences in implication between them, are one of the chief novel insights of the article. I think it would be useful to discuss how they might inform an understanding of LIN and LIN collapse as communication.

780-2: I don’t understand this sentence.

805: This figure is very confusing while also conveying little information. Everything it says could be communicated much more clearly with a t-chart, with the tops labeled “Favors early lysis” vs “favors late lysis” with descriptions below. (You could draw an arrow from the first to the second with a label “over time an infected culture may shift as a result of further infections”)

833: scares->scarce

861: Has it been shown that the collapse is actually adaptive for any virus? This is a really key question. (See below)

867: Here’s where a clear definition of communication really becomes key. Here’s one definition (Wilson, E.O. Sociobiology: The New Synthesis. Cambridge, MA: Harvard University Press (1975). “...the action of or cue given by one organism [the sender] is perceived by and thus alters the probability pattern of behavior in another organism [the receiver] in a fashion adaptive to either one or both of the participants.” By this definition, you need to show that the sender or receiver is getting a benefit. Your argument that LIN collapse is adaptive for the infected cell being lysed is mostly absent and is not very compelling (and this lack of evidence should be made clearer).

There is also a lack of empirical evidence that causing LIN collapse is adaptive. For mechanisms 1 and 2 described in Fig. 8, SA is not important, meaning that there is essentially by definition no communication required for these. And these are sufficient to cause LIN collapse.

Communication is, on the other hand, arguably involved in mechanisms 3 and 4, so perhaps synchronization involves communication. This would require LIN collapse to be adaptive for lysed cells, which I believe is plausible but not clearly demonstrated.

In any case, I think this involves some stretch to what we typically consider communication. For example, is SI a form of “coercive communication” that enforces cooperation of SA viruses to not infect the same cell and compete for resources? Is a fox eating a rabbit a form of “coercive communication” that enforces the rabbit’s cooperation in providing nutrients to the fox? Perhaps so, or perhaps LIN collapse is different, but I think you need to explore this.

This differs from LIN itself, which I think the evidence pretty clearly indicates is adaptive for the receiver and which doesn’t require the slippery slope of coercive communication.

921: I don’t really understand what this section adds to your paper.

941: This section seems conceptually linked to the earlier discussion of plaque formation. Merge and consolidate?

998: Not clear in text that this is all old ideas. In general, you should be more careful to delineate when you are presenting old ideas as opposed to new insights, especially as you have produced very many papers and chapters that often discuss similar ideas.

1003: I don’t believe you have shown LIN collapse to be adaptive.

Reviewer 3 Report

Minor specific comments:

Please correct typo in line 289 "chloroform-vaper treatment" In the Table 1 using an abbreviations to explain other terms may be a bit difficult for reader not familiar with the subject. Please use full names of the terms used to explain referred terms with abbreviation in parenthesis e.g.: instead of "SI (below) is "imposed upon homoimmune secondarily infecting phages...", please use: Secondary Infection (SI) is imposed upon homoimmune secondarily infecting phages..."

General comments:

I would love to see my results discussed more in this review. It may be not easy task, as they involve LIN players in non-LIN triggering environment (in most cases), but they show that RI and RIII may play also a non-LIN related functions and I postulate in my research that they may play much deeper regulatory functions. There is not that much evidence to support this claim, but I would really appreciate any discussion - also any argumentation against my hypothesis is very welcome! It would be more rewarding than simple statements to take look at my results (e.g. Golec et al.)

In general the review is very needed voice in discussion about old topic, which is slowly, as one of a mean of viral communication, getting back to the mainstream of virus research. 

Author Response

Minor specific comments:

Comment 3.1:

Please correct typo in line 289 "chloroform-vaper treatment"

Reply 3.1:

CORRECTED. THANK YOU!

Comment 3.2:

In the Table 1 using an abbreviations to explain other terms may be a bit difficult for reader not familiar with the subject. Please use full names of the terms used to explain referred terms with abbreviation in parenthesis e.g.: instead of "SI (below) is "imposed upon homoimmune secondarily infecting phages...", please use: Secondary Infection (SI) is imposed upon homoimmune secondarily infecting phages..."

Reply 3.2a:

I HAVE REDUCED RELIANCE ON ABBREVIATIONS IN THE TABLE, BUT WHILE STILL RETAINING EXTENSIVE REFERENCE TO THE ABBREVIATIONS SO AS TO ALLOW GREATER ASSOCIATION WITH THE ABBREVIATIONS AS USED IN THE MAIN TEXT AND, ESPECIALLY, FIGURES. I AM HOPEFUL THAT THIS APPROACH ADDRESSES THE DESIRES EXPRESSED IN THE COMMENT.

Reply 3.2b:

IN ADDITION, AS ABBREVIATION USE IN THE TABLE HAS AS A CONSEQUENCE OF THESE CHANGES BECOME SOMEWHAT REDUNDANT, I HAVE SEPARATED THE INDIVIDUAL ABBREVIATION ENTRIES FROM THE TABLE AND HAVE CREATED A PRIOR, ABBREVIATION-ONLY TABLE TO PROVIDE GUIDANCE SPECIFICALLY TO THE ABBREVIATION USAGE.

Comment 3.3:

General comments:

I would love to see my results discussed more in this review. It may be not easy task, as they involve LIN players in non-LIN triggering environment (in most cases), but they show that RI and RIII may play also a non-LIN related functions and I postulate in my research that they may play much deeper regulatory functions. There is not that much evidence to support this claim, but I would really appreciate any discussion - also any argumentation against my hypothesis is very welcome! It would be more rewarding than simple statements to take look at my results (e.g. Golec et al.)

Reply 3.3:

I AM ASSUMING THAT IT IS ESPECIALLY GOLEC ET AL. (2010) THAT IS BEING REFERRED TO HERE. I HAVE NOW ADDED TWO SUBSECTIONS (5.2.4 AND 5.3.2) TO THE MANUSCRIPT, TOTALING OVER 1000 WORDS, PLUS HAVE ADDED A SUMMARIZING TABLE. IN CAREFULLY READING THIS ARTICLE, IT HAS OCCURRED TO ME THAT THE GOLEC ET AL. RESULT, PARTICULARLY REGARDING THE EXTENSION OF LINed LATENT PERIODS IN THE PRESENCE OF EXCESS GENE rIII EXPRESSION, MIGHT BE SUPPORTIVE OF MECHANISM 1 OVER MECHANISM 2 (SUBSECTIONS 5.3.1 AND 5.3.2) AS UNDERLYING THE TIMING OF INITIATION OF LIN COLLAPSE. THANK YOU FOR THE CONTRIBUTIONS TO THIS INSIGHT!

Comment 3.4:

In general the review is very needed voice in discussion about old topic, which is slowly, as one of a mean of viral communication, getting back to the mainstream of virus research.

Reply 3.4:

I THANK THE REVIEWER FOR THEIR COMMENTS. THE MODIFICATION OF THE TABLE(S) WITH LUCK RESULTED IMPROVED READER APPRECIATION, AND THE NEW INSIGHT REGARDING THE RELEVANCE OF MECHANISM 2 WAS COMPLETE UNEXPECTED (TO ME).

Reviewer 4 Report

As a non-expert reader, I really enjoyed this review. It provides a very nice overview of the topic. This manuscript is very well written.

I’m really sorry for not being more helpful, but I really think this is a very nice piece of work!    

Author Response

As a non-expert reader, I really enjoyed this review. It provides a very nice overview of the topic. This manuscript is very well written.

I’m really sorry for not being more helpful, but I really think this is a very nice piece of work!

Reply 4.0:

I GREATLY APPRECIATE THE REVIEWER’S SUPPORT! YES, THERE ARE NOT A WHOLE LOT OF EXPERTS ON LYSIS INHIBITION LEFT IN THE WORLD, THOUGH MAYBE THAT MIGHT CHANGE JUST A LITTLE BIT WITH TIME. :-) I AM HOPING THAT THE NOW SUBSTANTIALLY REVISED MANUSCRIPT MAY BE SIMILARLY RECEIVED.